# Hypoxia induces senescence of bone marrow mesenchymal stem cells via altered gut microbiota

Junyue Xing[1,2,3], Yongquan Ying[4], Chenxi Mao[5], Yiwei Liu[1,2], Tingting Wang[1,2], Qian Zhao[1], Xiaoling Zhang[1,2], Fuxia Yan[3] & Hao Zhang[1,2,3]

Systemic chronic hypoxia is a feature of many diseases and may influence the communication between bone marrow (BM) and gut microbiota. Here we analyse patients with cyanotic congenital heart disease (CCHD) who are experiencing chronic hypoxia and characterize the association between bone marrow mesenchymal stem cells (BMSCs) and gut microbiome under systemic hypoxia. We observe premature senescence of BMSCs and abnormal D-galactose accumulation in patients with CCHD. The hypoxia that these patients experience results in an altered diversity of gut microbial communities, with a remarkable decrease in the number of *Lactobacilli* and a noticeable reduction in the amount of enzyme-degraded D-galactose. Replenishing chronic hypoxic rats with *Lactobacillus* reduced the accumulation of D-galactose and restored the deficient BMSCs. Together, our findings show that chronic hypoxia predisposes BMSCs to premature senescence, which may be due to gut dysbiosis and thus induced D-galactose accumulation.

[1] State Key Laboratory of Cardiovascular Diseases, Fuwai Hospital, National Center for Cardiovascular Diseases, Chinese Academy of Medical Sciences and Peking Union Medical College, Beijing 100037, China. [2] Key Laboratory of Cardiac Regenerative Medicine, National Healthy Commission, Fuwai Hospital, Beijing 100037, China. [3] Center for Pediatric Cardiac Surgery, Fuwai Hospital, CAMS&PUMC, Beijing 100037, China. [4] Department of Thoracic and Cardiovascular Surgery, Taizhou Hospital, Zhejiang 317000, China. [5] Department of Thoracic and Cardiovascular Surgery, 1st Affiliated Hospital of Wenzhou Medical University, Zhejiang 325006, China. These authors contributed equally: Junyue Xing, Yongquan Ying, Chenxi Mao. Correspondence and requests for materials should be addressed to H.Z. (email: Drzhanghao@yahoo.com)

Systematic chronic hypoxia is a pathological state associated with inadequate whole-body oxygen supply. A variety of diseases, including cyanotic congenital heart disease (CCHD), chronic obstructive pulmonary disease, chronic mountain sickness and pulmonary fibrosis, have been extensively reported to be implicated in this process[1, 2]. Clinically, the hypoxic condition concomitant with diseases not only exacerbates the process of disease per se but also brings about much more detrimental effects on the body[3]. Exploring the downstream pathophysiological responses to the systematic hypoxia markedly appears to be necessary for the disease cure, especially in the aspects of providing new therapeutic alternatives.

Many clinical studies have focussed on the capacity of bone marrow mesenchymal stem cells (BMSCs) to immunosuppress and tissue regeneration[4–6]. Notably, BMSC infusions have shown promise in ameliorating steroid-resistant acute graft-versus-host disease with prolonged overall survival after allogeneic stem cell transplantation[7–9]. Based on their potential to differentiate into various kinds of cells, BMSCs have been applied as tissue engineering patches/conduits in many clinical trials to treat spinal cord injury[10], ischaemic stroke[11, 12], myocardial infarction[13, 14], osteoarthritis[15] and other diseases. BMSCs are mainly located in the perivascular zone of the bone marrow (BM)[16]. Oxygen is a critical component of the BM niche, and a hypoxic environment in the BM is generally considered to be necessary for maintenance of normal physiological function and self-renewal of stem cells[17]. However, the hypoxic status in BM can be aggravated in patients experiencing systemic hypoxia and may induce distinctive physiological changes.

Mounting evidence has supported the hypothesis that communications exist between the gut and the BM[18, 19]. The gut microbiota has been shown to regulate haematopoietic stem cell differentiation by impairing BM function in obesity[20]. The gut lumen contains trillions of metabolically active microbes[21]. Both the microbiota and the highly vascularized intestinal mucosa that lies immediately adjacent to the gut lumen are sensitive to chronic hypoxia[22]. Therefore, the chronic profound hypoxia could alter the diversity of gut microbiota and result in the accumulation of gut microbiota-derived metabolites, which might significantly impact homeostasis of the host.

In this study, patients with CCHD were utilized as the human disease model of systemic chronic hypoxia to investigate the associations between the BM and gut. Intriguingly, we observe premature senescence of BMSCs in CCHD patients, which is implicated in gut dysbiosis and gut microbiota-derived D-galactose accumulation. Moreover, D-galactose accumulation in chronic hypoxic rats is decreased upon administration of moderate amount of *Lactobacillus* and also deficient BMSCs is restored significantly.

## Results

**Patient recruitment.** Ninety children undergoing surgical repair for CHD from 2013 to 2016 were enrolled. The exclusion criteria were as follows: age <6 months, diarrhoea, receiving antibiotics or hormone therapy within the last 12 weeks, extracardiac anomalies, and chromosomal abnormalities. The patients were divided into a non-cyanotic (NCy) group ($n = 45$) and a cyanotic (Cy) group ($n = 45$). All patient characteristics are summarized in Table 1. Forty samples were sent for metabolic profiling analysis ($n = 20$ per group), and another 40 samples were utilized for cell culture and phenotype analysis ($n = 20$ per group). Ten samples with adequate numbers of BM were randomly chosen for blood gas analysis ($n = 5$ per group). Among the 90 patients, 40 were chosen randomly for stool 16S ribosomal RNA and metagenomic sequencing ($n = 20$ per group).

**Table 1 Baseline characteristics of the enrolled patients**

|  | Group NCy ($n = 45$) | Group Cy ($n = 45$) | P |
|---|---|---|---|
| Age (months) | 23.10 ± 10.69 | 21.92 ± 18.78 | 0.217 |
| Male (%) | 48.89% | 53.3% | 0.673 |
| Height (cm) | 91.40 ± 6.33 | 78.33 ± 9.38 | <0.01 |
| Weight (kg) | 12.98 ± 1.69 | 9.61 ± 1.72 | <0.01 |
| Hb (g/dl) | 12.50 ± 0.95 | 16.91 ± 2.99 | <0.01 |
| Diagnosis | ASD12 VSD 33 | TOF 15 TGA 10 PAA 9 DORV 11 | — |

*ASD* atrial septal defect, *VSD* ventricular septal defect, *TOF* tetralogy of fallot, *TGA* transposition of the great arteries, *PAA* pulmonary artery atresia, *DORV* double outlet right ventricle

**Table 2 Arterial and BM blood gas analysis in the NCy and Cy groups**

|  | NCy group ($n = 5$) | Cy group ($n = 5$) | P |
|---|---|---|---|
| $SaO_2$ (%) | 99.13 ± 0.90 | 77.18 ± 5.89 | <0.01 |
| $PaO_2$ (mm Hg) | 115.90 ± 8.75 | 45.38 ± 5.13 | <0.01 |
| $SbmO_2$ (%) | 87.67 ± 3.06 | 43.00 ± 7.94 | <0.01 |
| $PbmO_2$ (mm Hg) | 51.67 ± 5.51 | 26.33 ± 3.05 | <0.01 |

*$SaO_2$* arterial oxygen saturation, *$PaO_2$* arterial blood oxygen partial pressure, *$SbmO_2$* bone marrow oxygen saturation, *$PbmO_2$* bone marrow oxygen partial pressure

**BMSCs from CCHD patients undergo premature senescence.** To evaluate the oxygen tension in BM and peripheral blood, the oxygen saturation of the patients with or without CCHD was measured. The partial pressure of oxygen ($pO_2$) and oxygen saturation ($SO_2$) were significantly reduced in the BM compared with the peripheral arterial blood in both groups (51.67 mm Hg vs. 115.9 mm Hg in the NCy group, $P < 0.01$; 26.33 mm Hg vs. 45.38 mm Hg in the Cy group, $P < 0.01$; Student's *t*-test). The $SO_2$ and $pO_2$ in the BM from the Cy group (26.33 mm Hg), however, demonstrated an even greater reduction (by half) than that in the BM from the NCy group (51.67 mm Hg, $P < 0.01$, Student's *t*-test) (Table 2).

To investigate the effect of chronic hypoxia on BMSCs, we cultured BMSCs and assessed the expression of a variety of senescence markers. The morphology and phenotype of BMSCs are shown in Supplementary Figure 1. Expression of p16 (~1.99 fold, $P = 0.002$, Student's *t*-test) but not p53, p21 or p27 was significantly higher in the Cy group than in the NCy group (Fig. 1a). The 3-[4,5-dimethylthiazol-2-yl]-2,5 diphenyl tetrazolium bromide (MTT) assay showed that BMSCs in the Cy group grew more slowly than those in the NCy group (Fig. 1b). The proportion of cells displaying the protein marker senescence-associated (SA)-β-galactosidase was dramatically increased in the Cy group (~57.95% vs. ~9.27% in the NCy group, $P < 0.01$, Student's *t*-test). The percentage of quiescent-state cells (G0/G1 phase) in the Cy group was significantly higher than that in the NCy group (~89.53% vs. ~74.59%, $P < 0.01$, Student's *t*-test), whereas the percentages of cells in the S and G2/M phases in the Cy group were much lower than those in the NCy group (Fig. 1d). The antiapoptosis assay showed a much greater proportion of BMSCs undergoing early apoptosis (Annexin V+/PI−) (~4.70% vs. ~1.80%, $P < 0.01$, Student's *t*-test) and late apoptosis (Annexin V+/PI +) (~6.50% vs. ~2.93%, $P < 0.01$, Student's *t*-test) in the Cy group than in the NCy group (Supplementary Fig. 1C),

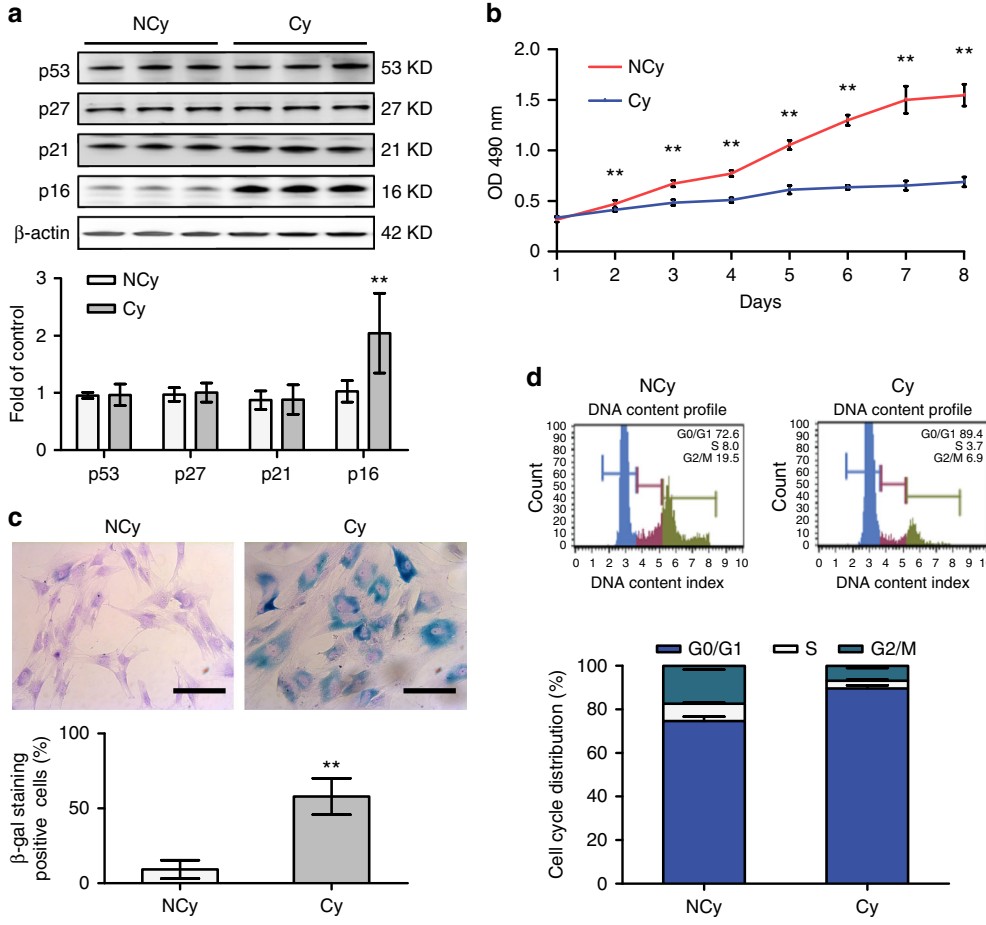

**Fig. 1** BMSCs from CCHD patients are predisposed to premature senescence. **a** BMSCs from non-CCHD (NCy) and CCHD (Cy) patients ($n = 5$ per group) were isolated and cultured to passage 3. Cell lysates were prepared to assess the levels of p53, p27, p21, p16 and β-actin by western blotting. **b** Cell numbers were determined by MTT assays at the indicated times. OD optical density. **c** SA-β-gal activity was assessed in the cells described in **a**. BMSCs stained in blue were labelled β-galactosidase-positive cells. Representative images are shown (upper). Scale bar, 50 μm. Twenty fields of each section were randomly selected to calculate the ratio of β-galactosidase-positive cells. **d** Cells described in **a** were collected and subjected to FACS analysis to assess the cell cycle distribution (upper). The percentage of cells in each cell cycle compartment is presented graphically (lower). Data represent the mean ± SD from three independent experiments, and statistical significance was analysed using Student's *t*-test (**$P < 0.01$)

indicating the attenuated antiapoptosis ability of BMSCs from patients with CCHD. Alizarin red staining and an alkaline phosphatase (ALP) activity assay (~0.10 ng/μl vs. ~0.32 ng/μl, $P < 0.01$, Student's *t*-test) revealed impaired osteogenesis potential in the Cy group (Supplementary Fig. 2A, D). In the adipogenic differentiation analysis, the differentiation rate in the Cy group was substantially lower than that in the NCy group (~11.37% vs. ~41.84%, $P < 0.01$, Student's *t*-test) (Supplementary Fig. 2B, E). In the chondrogenic differentiation assay (Supplementary Fig. 2 C, F), Type II collagen immunofluorescent staining showed a decreased differentiation rate in the Cy group (~30.84% vs. ~51.87%, $P < 0.01$, Student's *t*-test). Taken together, these results suggested that BMSCs from patients with CCHD were predisposed to premature senescence and had impaired multilineage differentiation potential.

**Profound hypoxia altered metabolic profiling in BM**. To explore the reason of BMSCs premature senescence in patients with CCHD, we employed a gas chromatography time-of-flight mass spectrometry (GC-TOF/MS)-based metabolomics approach to investigate metabolite alterations in the BM. In the total ion current profiles (Supplementary Fig. 3A), 543 valid peaks were identified. The partial least-squares discriminant analysis (PLS-DA) and orthogonal partial least-squares discriminant analysis

(OPLS-DA) exhibited a significant separation of clusters between the two groups (Supplementary Fig. 3B-D). A total of 49 metabolites displayed significantly different levels [variable importance in the projection (VIP, computes the influence on $Y$ of every term in the model, is the sum over all model dimensions of the contributions variable influence) >1, adjusted $P < 0.05$], as identified by the OPLS-DA model (Supplementary Table 1). The different metabolites were mainly composed of carbohydrates, amino acids, nucleotides and their intermediate metabolites. The metabolome map revealed that the different metabolites were enriched in glycolysis, gluconeogenesis, the TCA cycle and galactose metabolism pathways (pathway impact value >0.2, adjusted $P < 0.01$; Fig. 2a). These pathways are involved in glycometabolism and amino acid metabolism, which are required for energy production and protein synthesis, respectively.

Moreover, metabolomics highlighted D-galactose, a well-known agent for senescence induction through the production of large amounts of reactive oxygen species (ROS)[23], as the most relevant metabolite alteration in CCHD (47.07-fold). We also found an extremely high level of the D-galactose metabolic shunt metabolite galactonic acid (30.34-fold) and a low level of galactose-1-phosphate (0.433-fold). We tested the D-galactose in peripheral blood and fresh BM supernatant. D-Galactose levels in the Cy group remained higher than those in the NCy group, both in

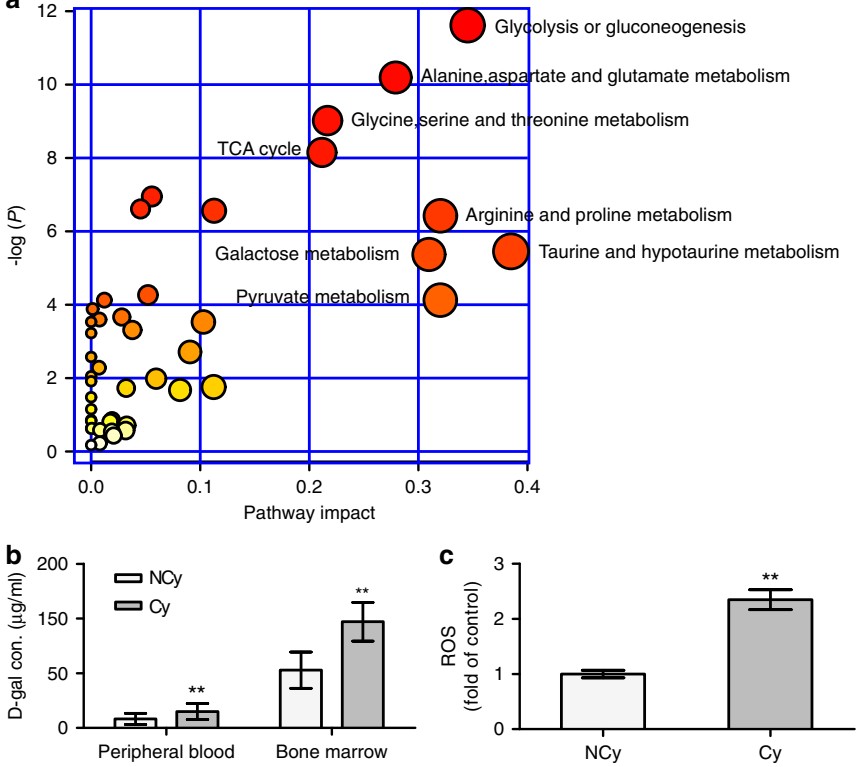

**Fig. 2** Profound hypoxia altered metabolic profiling in BM. **a** Pathway analysis of 49 metabolites identified as being present in the Cy group but not in the NCy group. x Axis represents the pathway impact, and y axis represents the pathway enrichment. Larger sizes and darker colours represent increased pathway enrichment and higher pathway impact values, respectively. **b** D-galactose concentrations in the peripheral blood ($n = 22$ per group) and bone marrow ($n = 5$ per group) of patients were analysed. **c** ROS levels of BMSCs from the NCy and Cy groups were analysed. The data shown are the mean ± SD from three independent experiments, and statistical significance was analysed using Student's t-test (**$P < 0.01$)

peripheral blood (~15.0 µg/ml vs. ~8.3 µg/ml, $P < 0.01$, Student's t-test) and in BM supernatant (~97.0 µg/ml vs. ~52.9 µg/ml, $P < 0.01$, Student's t-test) (Fig. 2b). Additionally, the Cy group showed a higher ROS level than the NCy group (~2.35-fold, $P < 0.01$, Student's t-test) (Fig. 2c).

**Accumulation of D-galactose induced BMSC senescence.** To investigate the association between D-galactose accumulation and premature senescence of BMSCs, BMSCs from the NCy group were cultured in an incubator with low (4%) or normal (21%) $O_2$ (designated Hypo- or Nor-), and the cells were exposed to normal (50 µg/ml) or high (90 µg/ml) concentrations of D-galactose (designated N- or H-). There was no difference between the Nor-N-Gal and Nor-H-Gal groups in terms of the expression of senescence-related genes (Fig. 3a). Compared with the Hypo-N-Gal group, the Hypo-H-Gal group presented 1.47-fold higher levels of P16. A two-way analysis of variance (ANOVA) showed that neither hypoxia ($P = 0.063$) nor D-galactose ($P = 0.181$) was a statistically significant factor for P16 expression, while interaction of these factors ($P = 0.004$, two-way ANOVA) had a significant effect on the level of P16. The levels of P53, P21 and P27 proteins and β-actin remained unchanged. Cell proliferation, SA-β-gal activity and multilineage differentiation were significantly attenuated only in the Hypo-H-Gal group (Fig. 3b, c; Supplementary Fig. 2). In the cell cycle assay, the number of cells in G0/G1 phase also increased in the Hypo-H-Gal group (Fig. 3d), indicating that the cell cycle was arrested by a synergistic combinatorial effect of high D-galactose and hypoxia. In addition, there was a distinct increase in ROS in the Hypo-H-Gal group (~1.70-fold) (Fig. 3e) compared with the Nor-N-Gal group. Our

findings indicated that D-galactose accumulation and hypoxia induced premature senescence of BMSCs in the NCy group.

**Profound hypoxia resulted in a reduction of *Lactobacillus*.** To elucidate the mechanism responsible for D-galactose accumulation, we identified the enzymes that mediate D-galactose metabolism in human BMSCs. The D-galactose metabolism pathway is shown in Supplementary figure 4 A. Western blot analysis showed no significant differences between the NCy and Cy groups in the levels of GALM, GALK1, GALK2, GLB1 and aldose reductase (Supplementary Fig. 4B).

Considering that lactose is the main source of D-galactose and is absorbed into the body via the intestine, 16S sequencing was performed and demonstrated marked alterations of the gut microbial communities in the Cy group (Fig. 4). The rarefaction analysis curves for each group were near saturation, suggesting that the sequencing data had a high quality and that very few new species were present (Fig. 4a). Analysis of similarity (ANOSIM) confirmed the significant separation of the groups ($R = 0.2$, $P = 0.001$, non-parametric analyses (ANOSIM)) (Fig. 4b), indicating clear differences in microbial composition in the NCy and Cy groups. The top 10 microbes at the phylum (Fig. 4c), family (Fig. 4d) and genus (Fig. 4e) levels are shown and indicate significant variations in the composition of the gut microbiota. Analyses of the microbiota at the phylum level revealed a dominance of *Firmicutes*, *Bacteroidetes* and Proteobacteria in both groups and a greater ratio of *Firmicutes/Bacteroidetes* (2.68-fold, $P = 0.001$, Student's t-test) in the Cy group than in the NCy group (Fig. 4c). We observed a decrease in the family *Lactobacillaceae* (0.05-fold, $P = 0.024$, metastats) and the genus *Lactobacillus* (0.05-fold, $P = 0.016$, metastats) in the Cy group (Fig. 4d–e). There was

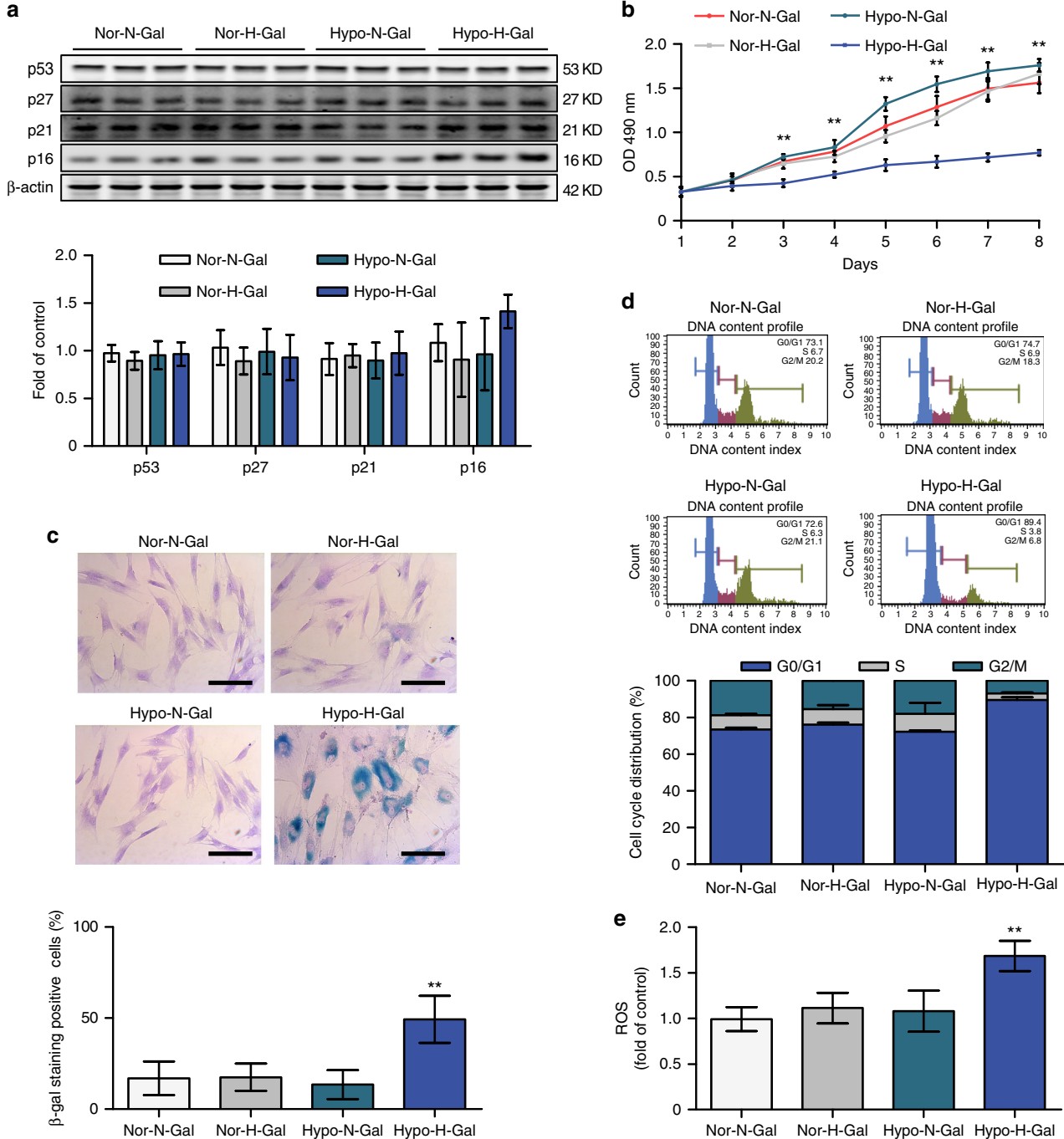

**Fig. 3** D-galactose accumulation and hypoxia led to premature senescence of BMSCs. **a** BMSCs from patients exposed to normal (50 µg/ml) or high (90 µg/ml) concentrations of D-galactose and cultured in a hypoxic (4% $O_2$) or normoxic (21% $O_2$) incubator. Samples were designated Hypo-N-Gal, Hypo-H-Gal, Nor-N-Gal or Nor-H-Gal accordingly. Cell lysates were subjected to western blot analysis to assess the protein levels of p53, p27, p21, p16 and β-actin. **b** Cell numbers were determined by the MTT assay. **c** The cells described in **a** were subjected to SA-β-gal analysis. Scale bar, 50 µm. **d** The cells described in **a** were subjected to FACS analysis. **e** Cellular ROS levels were analysed. The data shown are the mean ± SD from three independent experiments, and statistical significance was analysed by two-way ANOVA (**$P < 0.01$)

an increase in *Enterobacteriaceae* (2.06-fold, $P = 0.015$, metastats) but a decrease in *Lachnospira* (0.33-fold, $P = 0.045$, metastats), *Ruminococcaceae* (0.46-fold, $P = 0.014$, metastats) and *Faecalibacterium* (0.31-fold, $P = 0.014$, metastats) in the Cy group. LEfSe analysis showed an increased distribution of *Lactobacillus* in the NCy group (Fig. 4f). Furthermore, metagenomics sequencing showed that enzymes that degrade D-galactose (lactase (LCT), $P = 0.03$; aldose 1-epimerase, $P = 0.03$; GALE (UDP-glucose 4-

epimerase), $P = 0.01$; Fisher's exact test with false discovery rate (FDR) adjustment) were decreased in the Cy group (Fig. 4g).

Finally, Pearson correlation analyses (Fig. 4h) showed that the abundance of *Lactobacillus* in a patient's stool had a negative relationship with the D-galactose concentration in the patient's blood (Pearson correlation coefficients, $R = -0.706$, $P < 0.01$).

Collectively, patients with CCHD experienced gut microbiota dysbiosis, especially a reduction in *Lactobacillus*, which was

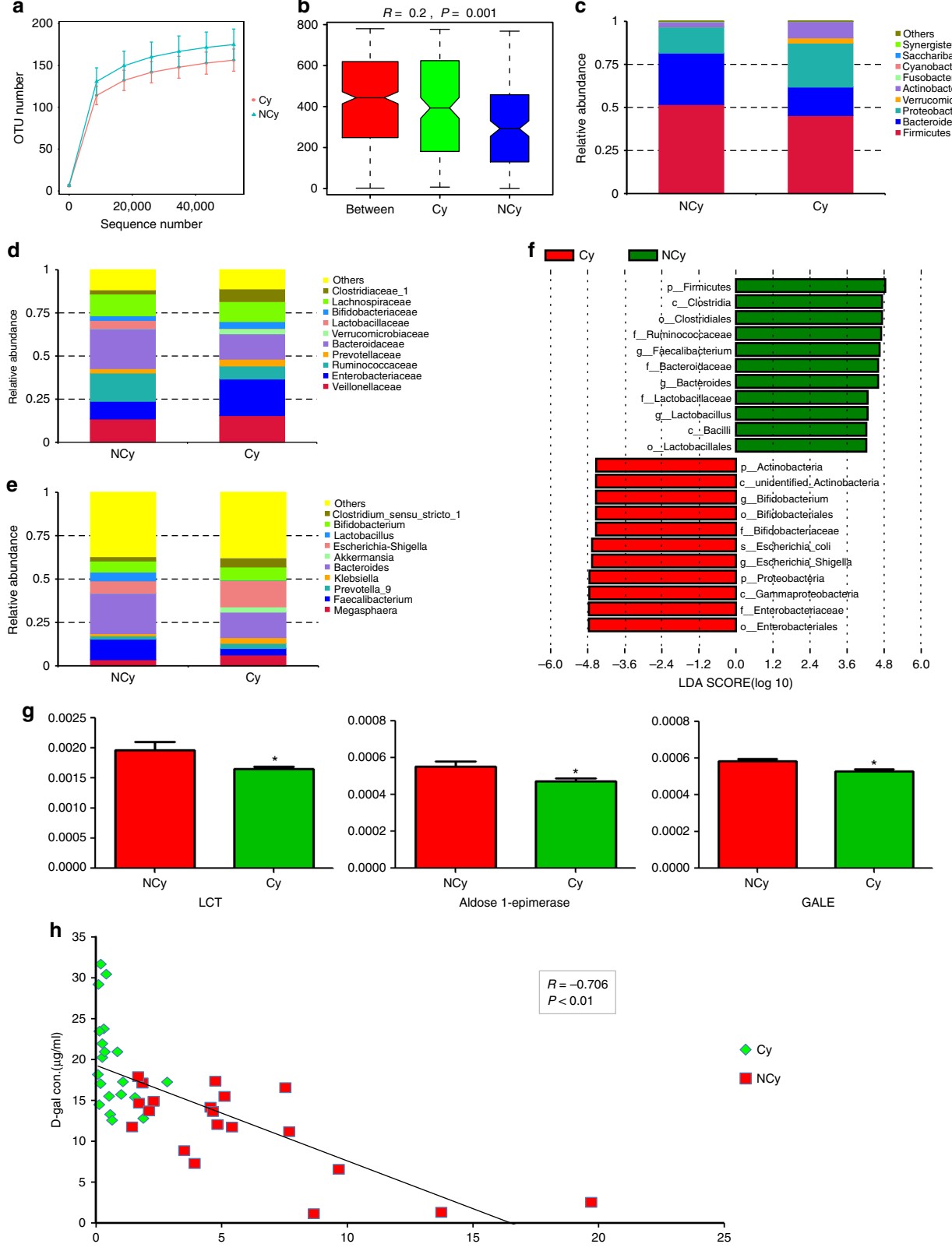

**Fig. 4** Profound hypoxia resulted in gut microbiota dysbiosis. **a** Rarefaction curves of the species number in the NCy and Cy groups (*n* = 20 per group). The curve in each group is nearly smooth when the sequencing data set is sufficiently large. **b** ANOSIM analysis of the beta diversity of the samples significantly separated the groups when *R* > 0 and *P* < 0.05. **c–e** The top 10 relative abundances of bacteria at phylum (**c**), family (**d**) and genus (**e**) levels in faecal samples from the NCy and Cy groups. **f** LEfSe analysis showed the association between two groups. **g** Metagenomic sequence analysis of the gene enrichment of enzymes that mediate D-galactose metabolism. LCT (*P* = 0.03), aldose 1-epimerase (*P* = 0.03), GALE (*P* = 0.01). **h** Pearson correlation analyses of *Lactobacillu*s in stool and D-galactose in blood (\**P* < 0.05)

negatively correlated with the D-galactose concentration in the patient's blood.

## D-galactose accumulation and BMSC dysfunction in hypoxic rats.

We established a chronic hypoxic rat model to confirm the effect of hypoxia on BMSCs. New-born Sprague-Dawley rats were housed in a hypoxic chamber under 10% $O_2$ or in ambient air for 3 weeks[24]. Haematocrit and haemoglobin levels in hypoxic rat blood were significantly higher than those in the normal group, and $pO_2$ and $SO_2$ levels were lower in the hypoxic group (Supplementary Fig. 5A), similar to the observations in patients with CCHD. The D-galactose concentration in rat blood (Fig. 5a), the *Lactobacillus* level in rat stool (Fig. 5b) and the expression of a variety of senescence markers (Fig. 5c) remained similar to those of the patients. D-galactose is primarily degraded by the liver, after being absorbed into the body via the intestine. We then tested D-galactose metabolism-related enzymes in rat liver. The expression of GALM/GALK2/GLB1/aldose reductase was unchanged, while GALK1 in rat livers decreased in the hypoxic group (Fig. 5d).

## *Lactobacillus* supplementation rescues deficient BMSCs.

The chronic hypoxic rats were intragastrically administered *Lactobacillus* ($1 \times 10^9$ cells per day)[25] or saline for the last week. Stool samples from rats that received the *Lactobacillus* showed increased levels of *Lactobacillus* (Fig. 6a). In rats that were not supplemented with *Lactobacillus*, the peripheral D-galactose concentration was higher under hypoxic conditions (in the hypoxia/− group) than under normal conditions (~2.88 µg/ml vs. ~1.25 µg/ml in the normal/− group, $P < 0.01$, one-way ANOVA followed by Tukey's test; Fig. 6b). When hypoxic rats were supplemented with *Lactobacillus* (hypoxia/+ group), the D-galactose concentration was substantially reduced (~1.55 µg/ml vs. ~2.88 µg/ml in the hypoxia/− group; $P < 0.01$, one-way ANOVA followed by Tukey's test) and very similar to the levels detected in rats housed under normal oxygen conditions without *Lactobacillus* supplementation (~1.55 µg/ml vs. ~1.25 µg/ml in the normal/− group, $P = 0.083$, one-way ANOVA followed by Tukey's test).

Haematoxylin and eosin (H&E) staining of the rat intestine revealed atrophy of the intestinal villus in the hypoxic group. *Lactobacillus* supplementation did not rescue this pathological change (Fig. 6c). The LCT activity in the rat intestine remained unchanged in all four groups ($P = 0.936$, one-way ANOVA) (Fig. 6d). However, *Lactobacillus* supplementation of the rats rescued their BMSCs from senescence (Fig. 6e, g), poor cell proliferation (Fig. 6f), cell cycle arrest (Fig. 6h), multilineage differentiation deficiency (Supplementary Fig. 6 A-F) and reduction in the liver GALK1 levels (Supplementary Fig. 6G).

To investigate whether the supplement of *Lactobacillus* provide long-term beneficial effects, rats were divided randomly into three groups and housed in the hypoxic chamber for 2 weeks, then supplied with *Lactobacillus* or saline for 3 weeks (−/− group, supplied with saline for entire 3 weeks; +/− group, *Lactobacillus* 1 week followed by saline 2 weeks; +/+ group, supplied with *Lactobacillus* for entire 3 weeks). Compared to the −/− group (Supplementary Fig. 7A), the *Lactobacillus* level in rat stool was increased significantly in the +/+ group (~3.19 fold, $P < 0.01$, one-way ANOVA followed by Tukey's test), and no significant difference was observed between the +/− and −/− group ($P = 0.334$, one-way ANOVA followed by Tukey's test). The D-galactose concentration (Supplementary Fig. 7B) in rat blood was

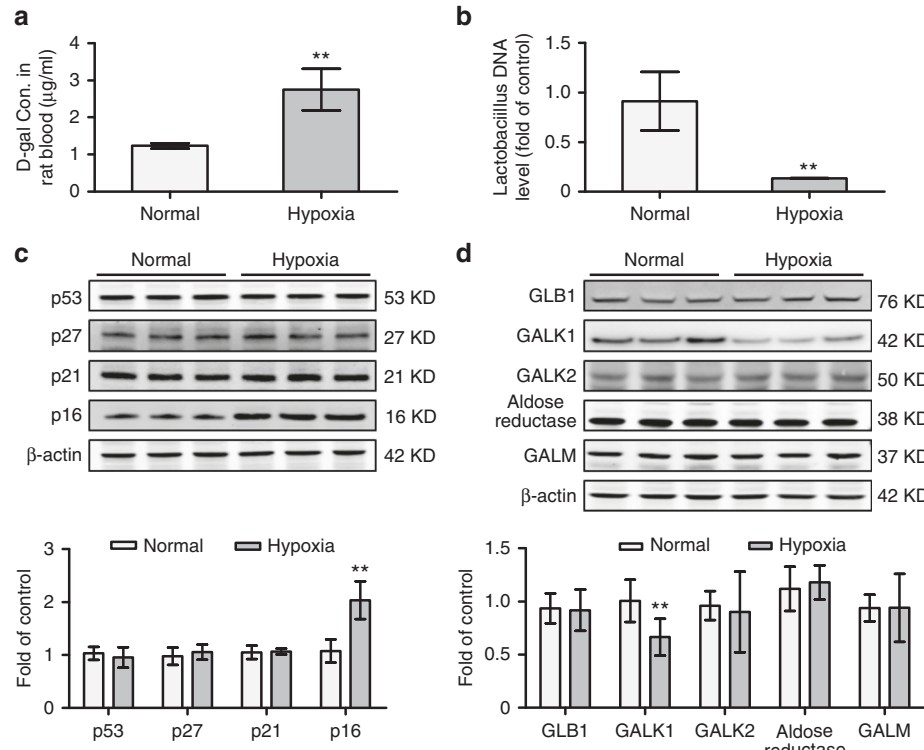

**Fig. 5** Accumulation of D-galactose and BMSC dysfunction in a rat chronic hypoxia model. **a** New-born Sprague-Dawley rats were housed in normoxic or hypoxic chambers for 3 weeks, respectively ($n = 9$ per group). Peripheral blood samples were obtained, and the D-galactose concentrations were measured. **b** Stools from the rats described in **a** were subjected to qPCR to assess the *Lactobacillus* content. **c** BMSCs from **a** were cultured to passage 3, and cell lysates were prepared to assess the levels of p53, p27, p21, p16 and β-actin by western blotting. **d** Western blotting was used to assess the protein levels of GALM, GALK1, GALK2, aldose reductase, GLB1 and β-actin in the rat liver. The data are the mean ± SD from three independent experiments, and statistical significance was analysed using Student's *t*-test (\*\**P* < 0.01)

lower in the +/+ group (~1.38 µg/ml vs. ~3.40 µg/ml in −/− group, $P < 0.01$, one-way ANOVA followed by Tukey's test) and back to higher level in the +/− group (~3.00 µg/ml vs. ~3.40 µg/ml in −/− group, $P = 0.164$, one-way ANOVA followed by

Tukey's test). The rescued BMSCs (as described above) showed a premature senescence phenotype in the +/− group, while continuous supplementation with *Lactobacillus* (+/+ group) show decreased proportion in SA-β-galactosidase (~20.31% in the

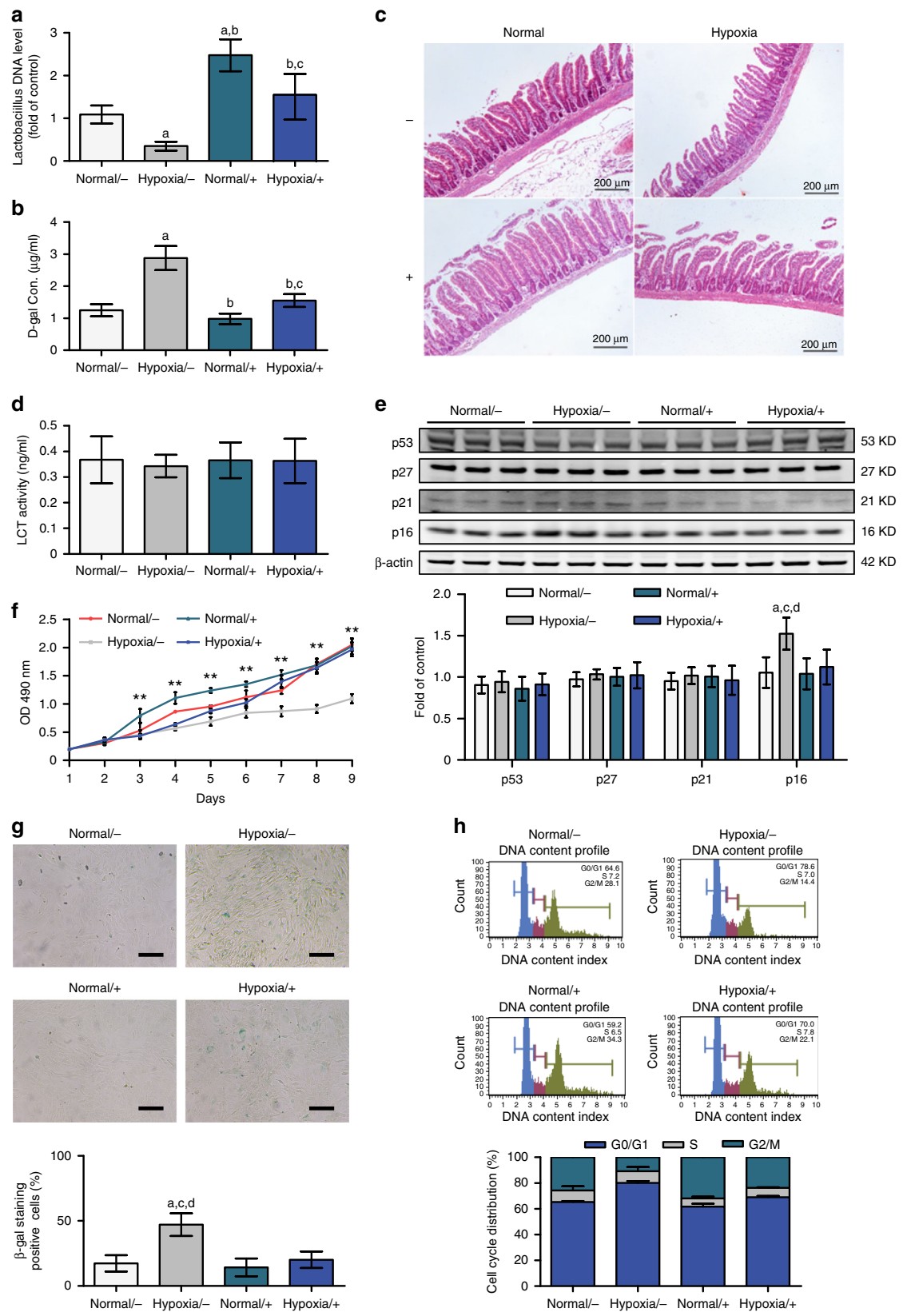

+/+ group vs. ~43.00% in the +/− group, $P = 0.013$, one-way ANOVA followed by Tukey's test; Supplementary Fig. 7C).

## Discussion

In the present study, we showed first that the chronic profound hypoxia experienced by patients with CCHD adversely influences BMSCs. Chronic profound hypoxia results in an altered diversity of gut microbial communities with a marked decrease in *Lactobacillus* and reduction of enzymes that degrade D-galactose. Cumulative D-galactose in the blood and BM induced BMSC premature senescence through ROS accumulation. Moreover, *Lactobacillus* supplementation in a rat model could recover the function of deficient BMSCs in the hypoxic rat model (Fig. 7).

Our results showed reduced oxygen tension in the BM of patients with CCHD than in NCy children. Low oxygen tension is essential for maintaining the pluripotency of stem cells[17], and the proliferation of BMSCs is enhanced when they are cultured in vitro under low pO₂ conditions (2–3% O₂)[26]. However, our results imply that the hypoxia-mediated protection of stemness is jeopardized in patients with CCHD. Previous studies have shown that various metabolites of cellular metabolism can affect BMSCs. It has been reported that 2,5-hexanedione can induce rat BMSC apoptosis[27] and that glyoxal can induce senescence in human BMSCs[28]. We explored metabolomics to elucidate distinctive BM metabolism signatures. Although no changes in 2,5-hexanedione and glyoxal were observed, we found that hypoxia in patients with CCHD altered the energy supply in BM and led to an accumulation of D-galactose.

As a constituent of milk, lactose is the major nutrient for infants and young children. Lactose is degraded to glucose and D-galactose in the intestine and absorbed into the body. Most of the absorbed D-galactose is degraded in the liver for energy production, while little is used for early development in humans[29]. D-galactose flows through the arteries to penetrate the cortical bone and medullary canal and then into the BM metaphyseal region and arterial capillaries. Based on the specified physiological structure of the BM, the peripheral blood and BM are well established for their interaction. Several studies have demonstrated that the metabolite distribution is significantly different but correlated between BM and peripheral blood[30, 31]. Our results showed that the D-galactose concentration was significantly increased in patients with CCHD, in both peripheral blood and BM. A series of enzymes that metabolize D-galactose are also present in BMSCs, but there were no significant differences between the NCy and Cy groups (Supplementary Fig. 4B). D-galactose has been extensively leveraged in mouse models of ageing[32, 33]. Therefore, the accumulated excessive D-galactose in BM could exert a negative effect on BMSCs.

High-dose D-galactose can induce premature ageing of astrocytes, foetal fibroblasts and neuroblasts[32, 34]. A number of reports have suggested that the ageing model induced by D-galactose is associated with oxidative stress[33]. Subsequent studies have demonstrated that ROS accumulation mainly accounts for this process[23]. Strikingly, although the D-galactose concentration

adopted in our in vitro cell culture study was far less than that used in mouse models of ageing, it still induced premature senescence of BMSCs under hypoxic conditions. Moreover, an increased level of ROS was detected only in BMSCs from patients with CCHD as well as the Hypo-H-Gal group. Therefore, profound hypoxia might be conducive to the function of D-galactose in downstream processes via alterations in energy production and ROS accumulation.

Intriguingly, BMSCs from patients with CCHD exhibited much worse antiapoptotic capability than those from NCy patients, indicative of the propensity for apoptosis in CCHD patients. Indeed, D-galactose has been reported to harbour a cytotoxic effect on cells, either in expediting senescence or in inducing apoptosis[35]. It is worth noting that both cytotoxic effects largely depend on ROS via the oxidative metabolism of D-galactose as well as advanced glycation end products[36]. It has been proposed that cells enter a state of senescence when the ROS level is sublethal[37] or apoptosis after exposure to high levels of ROS[38]. Although complete comprehension of the molecular mechanisms involved in cellular senescence and apoptosis remained to be attained, it is believed that the ROS level plays a pivotal role in balancing the two physiological processes[39]. Moreover, despite some controversies, ageing may predispose related cells to apoptosis by enhancing mitochondrial dysfunction[40, 41]. The mechanism responsible for this process clearly merits further characterization.

The gut microbiota had been reported extensively to be involved in lactose and D-galactose metabolism since the D-galactose metabolism pathway (Leloir pathway) is highly conserved and the enzymes in the Leloir pathway are also encoded by genes in *Lactobacillus*[42]. Lactose digestion can be improved by the modulation of colonic bacteria[43]. As human gut symbionts, the diversity of microbiota can, in general, be largely influenced by the diverse physiological states of the host[44]. We found that the Firmicutes/Bacteroides ratio increased in the Cy group, which is similar to findings obtained for patients with hypertension and obesity[19]. Enterobacteriaceae increased in the Cy group, whereas *Lactobacillus*, *Lachnospiraceae*, *Ruminococcaceae* and *Faecalibacterium* decreased. Previous studies have shown that many members of the *Enterobacteriaceae* family are pathogens, such as *Shigella*, one of the leading bacterial causes of diarrhoea[45, 46]. *Lachnospiraceae* may protect against colon cancer in humans by producing butyric acid[47]. Additionally, *Faecalibacterium* shows greater depletion in the gut microbiota of older heart failure patients[48]. In the present study, we noted that *Lactobacillus* markedly decreased in both patients with CCHD and in hypoxic rats. Intermittent hypoxia has been reported to alter the gut microbiota diversity in a mouse model of sleep apnoea, in which *Lactobacillus* does not appear to significantly change[49]. However, it has been noted that differing hypoxic status, whether intermittent or chronically profound, should be taken into account for commensal bacteria modulation. The gut microbiota is known to impact host distal organs via bioactive metabolites[19]. We found a strong negative correction between *Lactobacillus* reduction in the gut microbiota and D-galactose accumulation in the peripheral blood. Metagenomics sequencing of stool samples from children

**Fig. 6** Functional recovery of deficient BMSCs through *Lactobacillus* supplementation in the rat hypoxia model. **a** *Lactobacillus casei* CRL431 (+) or saline (−) was administered intragastrically to rats housed in normoxic or hypoxic chambers (*n* = 9 per group). DNA was extracted from stool samples and subjected to real-time qPCR analysis to assess the levels of *Lactobacillus* DNA. **b** The D-galactose concentration in peripheral blood samples was detected in the rats described in **a**. **c** Rat intestines were subjected to H&E staining. **d** Intestines from the rats described in **a** were subjected to ELISA of LCT activity. **e–h** BMSCs from rats described in **a** were isolated and cultured to passage 3. Cell lysates were analysed for ageing-related proteins by western blotting (**e**). Cells were subjected to MTT assays (**f**), SA-β-gal analysis (scale bar, 100 μm) (**g**) and FACS analysis (**h**) to assess cell growth, premature senescence status and cell cycle distribution, respectively. Data represent the means ± SD from three independent experiments, and statistical significance was analysed by one-way ANOVA followed by Tukey–Kramer multiple comparisons. $^{a}P < 0.05$ compared with normal/−. $^{b}P < 0.05$ compared with hypoxia/−. $^{c}P < 0.05$ compared with normal/+. $^{d}P < 0.05$ compared with hypoxia/+(**$P < 0.01$).

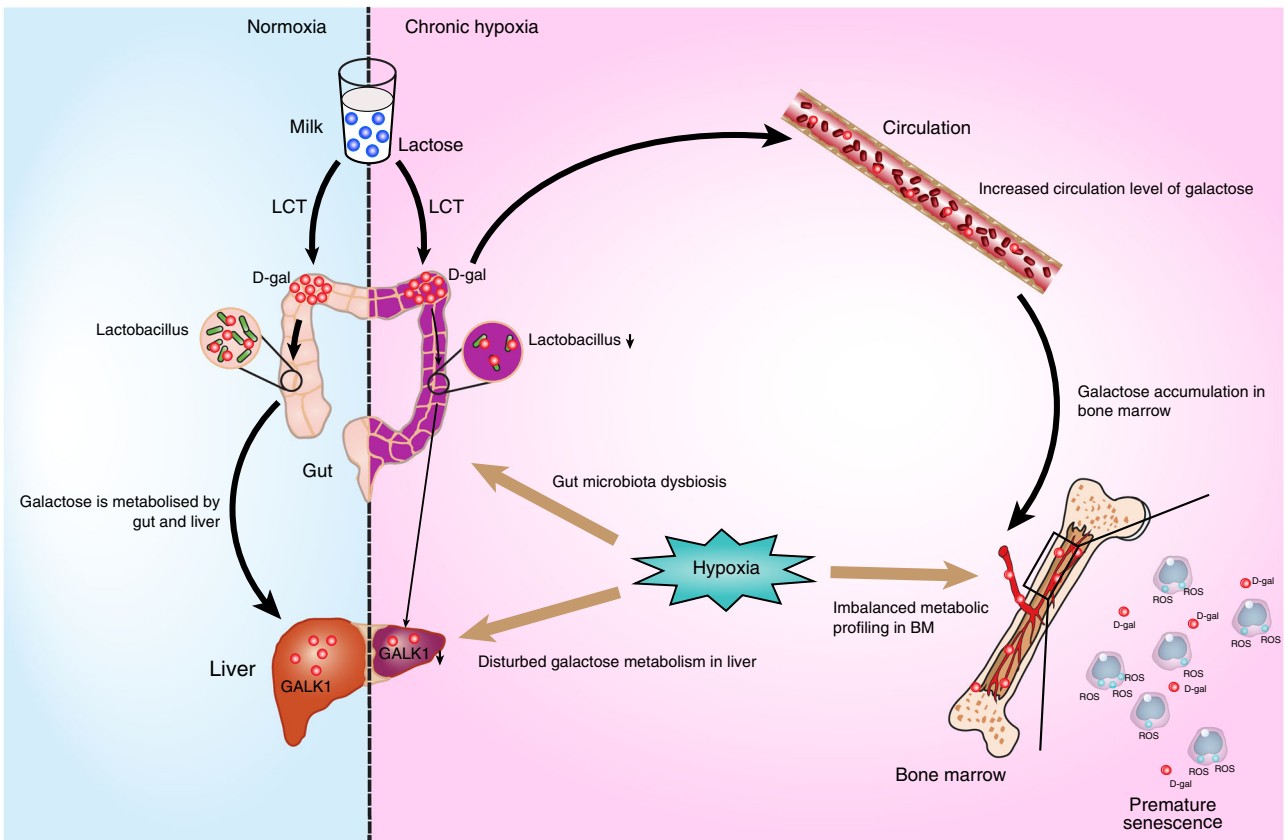

**Fig. 7** Summary Figure. As a constituent of milk, lactose is the major nutrient for infants and young children. Chronic profound hypoxia induced by CCHD results in alterations in the diversity of gut microbial communities, with markedly decreased *Lactobacillus* and a reduction in enzymes that degrade D-galactose. Cumulative D-galactose in the blood and BM induced premature senescence of BMSCs through ROS accumulation

revealed a distinct loss of D-galactose metabolism-related gene enrichment in the Cy group. These finding strongly indicated that the accumulation of D-galactose in the blood and BM could result from gut microbiota-derived metabolic alterations.

Lactose is also degraded by LCT, which is secreted by the intestinal gland around the brush border area. We did not observe impaired LCT activity in the intestine in the rat hypoxia model. Likewise, intestinal mucosa atrophy was noted in our model, which has been previously observed both in hypoxia rat[50] and patients with chronic obstructive pulmonary disease[51], indicating impaired mucosal barrier fortification and thus the entry of more bacterial metabolites into the circulation[50]. Given this case, we cannot exclude the fact that gut dysgenesis independent of microbiome is also a cause for the observed phenotype. We found that the replenishment of hypoxic rats with *Lactobacillus* could fully restore the deficient BMSCs, while the intestinal mucosa atrophy did not fully recover even after probiotic therapy. Therefore, the alterations of the gut microbial communities might be the main reason for the deficient BMSCs, and complete surgical repair of cardiac anomalies is warranted to relieve hypoxia and restore the integrity of the mucosal barrier.

*Lactobacillus* is one of the most important probiotic bacteria responsible for lactose and D-galactose metabolism in the intestinal tract. *Lactobacillus* is well used in microbial-mediated therapy. Although the atrophied intestinal villus with impaired mucosal barrier fortification and reduced GALK1 expression in the liver were observed in the rat hypoxia model, we found that the replenishment of hypoxic rats with *Lactobacillus* could reduce the accumulation of D-galactose and restore the deficient BMSCs. As a simple preventive approach, probiotics could produce antimicrobial substances, prevent pathogen and toxin adsorption,

and modulate the immune response[52], which has been applied in some clinical studies to prevent necrotizing enterocolitis among the neonatal and paediatric population[22]. In the clinical setting, oral probiotic supplementation is more convenient than faecal microbiota transplantation, especially for children with CCHD. However, when *Lactobacillus* was removed from the diet of hypoxic rats, the BMSCs showed premature senescence again. It strongly suggested that continuous supplementation was necessary before the correction of hypoxia status.

Undoubtedly, our study has some limitations. First, various diseases are known to be involved in systematic chronic hypoxia. The disease model for this study confined merely to CCHD patients obviously appears to be incomplete. Second, the patients enrolled in the present study were young children. A recent study showed a link between *Lactobacillus* and blood pressure among healthy adults[53]. The effect of hypoxia and gut microbiota remains to be further characterized in adults. Third, given that BMSCs and haematopoietic stem cells (HSCs) share the same microenvironment, it is conceivable that the accumulated D-galactose in CCHD patients likely influenced the function of HSCs and even other progenitor cells residing in the bone marrow. Recently, Randa et al.[54] has suggested that the increased immature platelet fraction in CCHD patients may denote peripheral platelet destruction. Apparently, haematopoiesis is perturbed to some extent in CCHD patients. However, the effect of hypoxia driven by CCHD on HSCs remains far to be determined owing to the lack of related studies. The dynamic change in HSCs and likely malfunction of haematopoiesis in CCHD patients requires further study.

In summary, BMSCs from patients with CCHD were predisposed to premature senescence, which might have resulted from gut dysbiosis and gut microbiota-derived D-galactose

accumulation. *Lactobacillus* supplementation restored the deficient BMSCs. Our findings suggested that oral *Lactobacillus* supplementation might enhance the efficiency of autologous BMSC-based regenerative therapy and open possibilities to improve stem cell-based cures for patients with CCHD.

## Methods

**Study approval.** This study was approved by the ethics committee of Fuwai Hospital, and the study protocol conformed to the ethical guidelines of the 1975 Declaration of Helsinki. The parents of all enrolled patients provided informed consent.

All animal procedures were conducted in accordance with the procedures reviewed and approved by the Care of Experimental Animals Committee. All efforts were made to minimize animal suffering.

**Patients.** Patients undergoing surgical repair for CHD were intubated and supplemented with 21% oxygen after induction of general anaesthesia in the operating room. Next, 1–2 ml of BM was aspirated from the sternum of the 90 patients undergoing open-chest cardiac surgery. The samples were used for metabolic profiling, cell culture and phenotype and blood gas analyses as described.

**Rat.** A rat model was established to mimic the chronic hypoxic status experienced by patients with CCHD[24]. Briefly, new-born Sprague-Dawley rats were divided randomly into normal and hypoxic groups. The hypoxic group of rats were housed in a hypoxic chamber (ProOx110, Biospherix, NY) under 10% $O_2$ for 3 weeks, whereas rats in the normal group were housed in ambient air for the same period.

For supplementation with *Lactobacillus*, new-born Sprague-Dawley male rats were divided randomly into four groups ($n = 9$ per group); two groups were housed in a normal environment, while the other two were housed in a 10% $O_2$ chamber for 3 weeks. One group in the normoxic chamber (normal/+ group) and one group in the hypoxic chamber (hypoxia/+ group) were intragastrically administered *Lactobacillus casei* CRL431 (Chr. Hansen A/S, Horsholm) for the last week ($1 \times 10^9$ cells per day)[25], while the other groups in the normoxic (normal/− group) and hypoxic (hypoxia/− group) chambers received saline intragastrically. Rat blood, stool, intestine and BMSCs were obtained and analysed.

To check whether the *Lactobacillus* effects on hypoxic rats are permanent or simply acute, new-born Sprague-Dawley male rats were divided randomly into three groups ($n = 6$ per group) and housed in the hypoxic chamber. Two weeks later, two of them were supplied with *Lactobacillus* (+/+ group and +/− group) for 1 week, while the −/− group was supplied with saline (−/−). Then the +/+ group was supplied with *Lactobacillus* for additional 2 weeks, and the rat in the +/− and −/− groups received saline instead. Rat blood, stool and BMSCs were obtained and analysed.

**Cell culture and phenotype analysis.** Human and rat BMSCs were isolated and cultured in Dulbecco's Modified Eagle's Medium (DMEM, HyClone) supplemented with 10% foetal bovine serum (FBS) and antibiotics at 37 °C in 5% $CO_2$. The cell morphology was observed with an inverted microscope. Cell surface markers were measured with the Human MSC Analysis Kit (BD, Biosciences, San Jose, CA) by following the manufacturer's instructions. Third-passage BMSCs were cultured. CD90, CD105, CD73, CD44, CD34, CD11b, CD19, CD45 and HLA-DR were all tested by flow cytometry for the phenotypic analysis.

To evaluate the effect of D-galactose on BMSCs, BMSCs from children in the NCy group were cultured in an incubator with low (4%) or normal (21%) $O_2$, and the cells were exposed to normal (50 µg/ml) or high (90 µg/ml) concentrations of D-galactose (Sigma, St. Louis, MO). These cells were designated Hypo-N-Gal, Hypo-H-Gal, Nor-N-Gal and Nor-H-Gal, respectively. After 21 days, the BMSCs were harvested for β-galactosidase staining and the MTT and cell cycle assays.

**Antibodies and western blot analysis.** Cell lysate was extracted by RIPA buffer with protease inhibitors (Roche). Protein samples were run on sodium dodecyl sulfate-polyacrylamide gel and transferred onto nitrocellulose membranes by an iBlot 2 dry blotting system (Thermo Fisher Scientific). Membranes were blocked in 5% non-fat dry milk in TBST (100 mmol/l Tris, pH 7.5, 0.9% NaCl, 0.1% Tween-20) for 1 h and incubated with primary antibodies overnight at 4 °C. Membranes were washed with TBST 3 times and incubated for 1 h with Dylight 800 goat anti-mouse-IgG (1:2000, EarthOx, E032810) or Dylight 800 goat anti-rabbit-IgG secondary antibody (1:2000, EarthOx, E032820) at room temperature. Signals were detected by using Odyssey CLx Western Blot Detection System.

The senescence-related gene expression levels in passage-3 BMSCs were analysed. Monoclonal anti-P16 (1:1000, Abcam, Cat# ab108349), anti-P53 (1:1000, Abcam, Cat# ab179477), anti-P27 (1:1000, Abcam, Cat# ab32034), anti-P21 (1:1000, Abcam, Cat# ab109520) and anti-β-actin (1:10000, Proteintech, Cat# 60008-1-Ig) were used.

D-gal metabolism-related enzymes were analysed in human BMSCs, rat BMSCs and rat liver. Monoclonal anti-GLB1 (1:50000, Abcam, Cat# ab128993), anti-aldose reductase (1:100, Santa Cruz, Cat# sc-373953) and anti-GLAM (1:100, Santa Cruz, Cat# sc-166304) and polyclonal anti-GALK1 (1:1000, abcam, Cat# ab65123) and anti-GALK2 (1:1000, abcam, Cat# ab153815) were used.

The uncropped western blots and all the western blot repeats (at least three replicates) were included in Supplementary Figure 8 and Supplementary Figure 9, respectively.

**β-Galactosidase staining.** BMSCs were cultured to passage 3 and plated in 6-well plates for β-galactosidase staining following the manufacturer's instructions provided with the β-Galactosidase Staining Kit (Cell Signaling Technology, Danvers, MA). Cells were washed in phosphate-buffered saline (PBS) and fixed for 5 min at room temperature in 4% formaldehyde. Then cells were incubated overnight (without $CO_2$) at 37 °C with SA-β-gal stain solution (1 mg/ml X-gal, 40 mM citric acid/sodium phosphate, pH 6.0, 5 mM potassium ferrocyanide, 5 mM potassium ferricyanide, 150 mM NaCl, 2 mM $MgCl_2$). Images were collected using a DM750 microscope (Leica, Germany). Twenty fields of each section were randomly selected to calculate the ratio of β-galactosidase-positive cells.

**MTT assays.** Cells were plated in 96-well plates at a density of $2.0 \times 10^3$ cells per well for the MTT assay. The optical density at 490 nm was determined for each well using an absorbance plate reader from the first day to the eighth day, and the growth curves were plotted to estimate the proliferation potential of BMSCs from various groups.

**Cell cycle assays.** For the cell cycle assay, BMSCs were plated in 6-well plates and treated with serum-free medium for 12 h to achieve cell synchronization. Next, DMEM supplemented with 10% FBS was added to promote BMSC division. After 24 h, BMSCs were collected, incubated in 37 °C with RNase A (20 µg/ml; Thermo) for 30 min and then stained with propidium iodide (PI; 50 µg/ml) (Beyotime, China). Cell cycle analysis was performed using BD accuri C6.

**Antiapoptosis assays.** BMSCs were cultured with serum-free DMEM in a hypoxia incubator for 12 h to induce cell apoptosis as previously described[55]. The cells were then collected and incubated with the fluorescent dye Annexin V and PI. Apoptosis was measured by flow cytometry with a BD accuri C6.

**Multilineage differentiation potential assay of BMSCs.** For osteogenic differentiation, BMSCs were seeded at $2 \times 10^4$/cm in 6-well plates. In all, 0.1 mM dexamethasone, 10 mM β-glycerol phosphate and 50 mM ascorbate were added to the medium to induce differentiation[56]. The medium was changed every 3 days. Seven days later, ALP activity were measured by the Alkaline Phosphatase Detection Kit (Sigma) according to the manufacturer's instructions. And 3 weeks later, cells were fixed in 4% paraformaldehyde for 15 min, washed 3 times with PBS and then stained with Alizarin red (1% in 0.1 M Tris-HCl (pH = 8.3)) for 5 min to assess the degree of osteogenic differentiation,

BMSCs were seeded at $2 \times 10^4$/cm in 6-well plates. Adipogenic differentiation was induced in the medium with 1 µM dexamethasone, 200 µM indomethacin, 0.5 µM 1-methyl-3-isobutylxanthine and 10 µg/ml insulin[57]. Three weeks later, cells were fixed in 4% paraformaldehyde for 15 min, washed 3 times with PBS, then stained with (3 g/l) Oil Red O for 30 min, washed in 75% ethanol and stained with haematoxylin for 1 min to assess adipogenic differentiation by microscope DM750 (Leica, Germany). Twenty fields of each section were randomly selected to calculate the ratio of adipogenic differentiation.

To induce chondrogenic differentiation, BMSCs were gently centrifuged($250 \times g$, 10 min) to form a pelleted micro-mass and cultured with 0.1 µM dexamethasone, 0.17 mM ascorbate, 10 µg/l TGF-β3, ITS+supplement, 1 mM sodium pyruvate and 0.35 mM proline[57]. Three weeks later, cell pellets were fixed in 4% paraformaldehyde for 30 min and dehydrated through an alcohol gradient series, embedded in paraffin wax blocks. Before staining, 2-µm-thick sections were dewaxed in xylene, rehydrated through decreasing concentrations of ethanol, blocked in 5% goat serum (with 0.3% Triton X-100) for 1 h at 37 °C and incubated with Collagenase II antibody (1:250, Santa Cruz, Cat# sc-52658) overnight at 4 °C. Washed 3 times with PBS and incubated with FITC-labelled goat anti-mouse-IgG (1:1000, ZSGB-Bio, Cat# ZDR-5210) at room temperature for 1 h. Images were collected using a DM750 microscope (Leica, Germany). Twenty fields of each section were randomly selected to calculate the ratio of chondrogenic differentiation.

All of the reagents used in multilineage differentiation assay are from Sigma.

**Metabolic profiling analysis.** BM supernatants were collected from the patients after centrifugation. Next, 350 µl methanol was added to 100 µl BM supernatants containing 50 µl of L-2-chlorophenylalanine as an internal standard. The lyophilized metabolite extract was then sequentially oximated with methoxyamine hydrochloride and silylated with BSTFA reagent (1% TMCS, v/v; REGIS Technologies, USA). Metabolomic analysis was performed using an Agilent 7890 gas chromatograph system coupled to a Pegasus HT time-of-flight mass spectrometer (GC-TOF/MS, LECO, St. Joseph, MI, USA).

Chroma TOF4.3× software (LECO, St. Joseph, MI, USA) was used for raw data analysis, including raw peak extraction, baseline data filtering, calibration of the baseline, peak alignment, deconvolution analysis, peak identification and integration of the peak area. The SIMCA-P 13.0 software package (Umetrics, Umea, Sweden) was then used for principal component analysis (PCA), PLS-DA and OPLS-DA. A leave-

one-out validation was used to estimate the robustness and predictive ability of the mode. Metabolites that were significantly different between the NCy and Cy groups were screened using the OPLS-DA model. The principal component of the VIP >1.0 was first selected as a changed metabolite. The nonparametric univariate method (Mann–Whitney–Wilcoxon test) was used to analyse metabolites that differed in abundance between the NCy group and the Cy group corrected for the FDR to ensure that the peak of each metabolite was reproducibly detected in the samples. The metabolites detected using GC-TOF/MS were identified by searching against the LECO/Fiehn Metabolomics Library and validated with standard substances if available. The Kyoto Encyclopaedia of Genes and Genomes (http://www.genome.jp/kegg/) and MetaboAnalyst 3.0 (http://www.metaboanalyst.ca/MetaboAnalyst/) were utilized to analyse the metabolic pathways.

**Analysis of oxygen concentration and D-galactose in blood and BM.** Blood gas analysis for the freshly harvested BM was performed according to a previously described protocol[58]. Briefly, 1–2 ml of whole BM marrow or blood was aspirated into a blood gas syringe. The $PO_2$ and $SO_2$ were measured by blood gas analyser CCX1 (NOVA, USA).

Blood samples (1–2 ml) were obtained and allowed to stand for 30 min, centrifuged at $1000 \times g$ for 10 min at room temperature and then transferred the supernatant to liquid nitrogen immediately. All samples were stored at $-80\,°C$ until use.

A Galactose Colorimetric/Fluorometric Assay Kit (Sigma, St. Louis, MO) was used to measure D-galactose in the BM supernatant and peripheral blood according to the manufacturer's instructions in both the human and rat. In brief, 50 μl serum were incubated with 50 μl Master Reaction Mix (44 μl Galactose Assay Buffer, 2 μl Galactose Probe, 2 μl Galactose Enzyme Mix, 2 μl HRP) for 30 min at 37 °C in dark. The absorbance at 570 nm were measured.

**Stool 16S ribosomal RNA sequencing and metagenomics sequencing.** Forty patients were randomly chosen for stool 16S ribosomal RNA and metagenomics sequencing ($n = 20$ per group). Fresh faecal samples were collected before cardiac surgery. The inside portions of the stool were frozen in liquid nitrogen within 1 h of collection. All samples were stored at $-80\,°C$ until use.

Bacterial genomic DNA from stools was isolated using a QIAamp DNA Stool Mini Kit (QIAGEN). The 16S ribosomal RNA sequencing and metagenomics sequencing were used to analyse differences between the NCy and Cy groups on the Illumina platform at Novogene Bioinformatics Technology Co., Ltd. The 16S rRNA genes of distinct regions (V3–V4) were amplified using specific primers (341F: CCTAYGGGRBGCASCAG; 806R: GGACTACNNGGGTATCTAAT) Sequencing libraries were generated using the TruSeq® DNA PCR-Free Sample Preparation Kit (Illumina, USA). The library was sequenced on an Illumina HiSeq2500 platform, and 250-bp paired-end reads were generated. Sequence analyses were performed using the Uparse software. Sequences with ≥97% similarity were assigned to the same operational taxonomic units (OTUs). Additionally, based on the normalized OTUs, alpha diversity was applied to analyse the complexity of species diversity for the samples. ANOISM analysis was used to evaluate differences among the samples in species complexity.

For metagenomics sequencing, a total of 1 μg of DNA per sample was used as input material for the DNA sample preparations. Sequencing libraries were generated using the NEBNext® Ultra™ DNA Library Prep Kit for Illumina (NEB, USA) according to the manufacturer's recommendations, and index codes were added to attribute the sequences to each sample. Briefly, the DNA sample was fragmented by sonication to a size of 350 bp, and then the DNA fragments were end-polished, A-tailed and ligated with the full-length adaptor for Illumina sequencing for further PCR amplification. Finally, the PCR products were purified (AMPure XP system), and the libraries were analysed for size distribution using an Agilent2100 Bioanalyser and quantified using real-time PCR. Clustering of the index-coded samples was performed on a cBot Cluster Generation System according to the manufacturer's instructions. After cluster generation, the library preparations were sequenced on an Illumina HiSeq platform, and paired-end reads were generated.

Real-time PCR was performed to identify changes in the bacterium in rat and human stool. The following primer pairs were used: Lactobacillus-1 F: 5′-AGCAGTAGGGAATCTTCCA-3′, Lactobacillus-1 R: 5′-CACCGCTACACATGGAG-3′, Lactobacillus-2 F: 5′-GCCTTGTACACACCGCCCGT-3′, Lactobacillus-2 R: 5′-CTCAAAACTAAACAAAGTTTC-3′, Universal bacterial primer F: 5′-CCTACGGGAGGCAGCAG-3′, and Universal bacterial primer R: 5′-ATTACCGCGGCTGCTGG-3′.

**Histology of the intestine and LCT activity analysis.** Rat intestines were immersed in 4% paraformaldehyde for 4 h and transferred to 70% ethanol. Biopsy material was placed in processing cassettes, dehydrated through an alcohol gradient series and embedded in paraffin wax blocks. Before staining, 5-μm-thick tissue sections were dewaxed in xylene, rehydrated through decreasing concentrations of ethanol, washed in PBS and stained with H&E. After staining, the sections were dehydrated in increasing concentrations of ethanol and xylene. Images were collected using a DM750 microscope (Leica, Germany). Rat intestine was obtained for LCT activity analysis using a Rat LCT ELISA Kit (Cusabio, Wuhan, China).

**Statistics.** Statistical analysis was performed using the SPSS 16.0 software (SPSS, Inc., Chicago, IL, USA). Normality of distributions were tested by Shapiro–Wilk test. Homogeneity of variance was determined by Levene's test. Differences between groups were examined using Student's $t$-test, chi-square test and one-way ANOVA followed by post hoc Tukey test, two-way ANOVA.

PCA, PLS-DA and OPLS-DA were used for metabolic profiling analysis. The Mann–Whitney–Wilcoxon test was used to analyse metabolites that differed in abundance between groups corrected for the FDR. The $P$-values from the enrichment analysis were adjusted for multiple testing of FDR.

The Wilcoxon rank-sum test, LefSe, ANOSIM and Metastats[59] were used for the 16S ribosomal RNA and metagenomic sequencing analyses. The different abundance of microbiota and enzymes between groups were analysed by Fisher's exact test with FDR adjustment. Correlations between the contents of Lactobacillus and D-galactose were identified by Pearson correlation analysis.

All $P$-values were two-sided, and $P < 0.05$ was considered statistically significant. All data are expressed as mean ± SD.

**Data availability.** All data generated or analysed during this study are available from the authors upon reasonable request. All 16S rRNA sequences have been deposited to SRA database under the accession codes : SRP140781.

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

## Acknowledgements

This study was supported by the National Science Fund for Distinguished Young Scholars (81525002), National Natural Science Foundation of China (81430006 and 81400186) and Professor in Peking Union Medical College.

## Author contributions

H.Z. conceived and supervised the study. Y.Y., Z.M., T.W. and F.Y. collected the patient samples. J.X., Y.Y. and M.X. performed the data analysis for metabolomics, 16S ribosomal RNA sequencing and metagenomic sequencing. J.X., Y.Y. and M.X. performed the cell study. J.X. and Y.L. performed the animal study. J.X., Y.Y., C.M. and Q.Z. prepared the manuscript draft. J.X. and H.Z. revised the manuscript and provided extensive discussions. All authors participated in discussion and editing of the manuscript.

## Additional information

**Competing interests:** The authors declare no competing interests.

