## [Peer Review File · Nature Communications]

Reviewers' comments:

Reviewer #1 (Remarks to the Author):

The manuscript by Xing et al. describes the hypoxia-induced changes to gut microbiota and bone marrow MSCs. They conclude that severe hypoxia induces gut microbiota dysbiosis and that this in turn leads to higher levels of D-galactose and subsequent MSC dysfunction and senescence. Using both patient data and a rat model, they suggest that introducing *Lactobacillus* can reduce D-galactose levels and rescue MSC function. The data are presented clearly and represent a thorough analysis of both human samples and rat tissue. The overall observations are very intriguing, but some conclusions seem over-interpreted, based on the data provided. It is also unclear from the manuscript why MSC function is the focus or read-out. The authors reference two papers suggesting that bone marrow stem cells can be used to treat myocardial infarction, but it seems very loosely connected to this study. Bone marrow contains hematopoietic, endothelial, and mesenchymal stem cells.

Major.

1. It is not clear how *Lactobacillus* would regulate D-gal in the bone marrow. There are much higher levels of D-gal in the bone marrow- even non-Cy, as compared to the peripheral blood. What is the connection between the intestine and where *Lactobacillus*, presumably, is located, and the bone marrow? This is not clearly explained or discussed.
2. Are there other reasons why there might be elevated D-gal concentrations in CCHD patients? Is D-gal on apoptotic cells? Are there more apoptotic cells in these patients?
3. One disconnect is why BM MSCs are important in heart disease? Are there confounding impacts of CCHD on hematopoiesis, for example?
4. Legends in some figures are almost impossible to read due to the very small font size (Figure 4, for example).

Minor

1. The entire manuscript contained grammatical mistakes. Please review carefully.
2. The histograms are not labeled clearly in Supp. Figure 1.
3. The first part of the results section where patient enrollment is described is confusing- particularly the last statement which states "among them"- it is not clear who is "them".

Reviewer #2 (Remarks to the Author):

The study by Zhang and colleagues presents a novel angle that could be involved in cyanotic congenital heart disease, proposing a hypoxia-gut-bone marrow link and feedback loop. They propose that hypoxia because of a mal-functioning heart modulates the gut microbiota and reduces *Lactobacillus*, thus resulting in an accumulation of D-galactose. This in turn drives cells in the bone marrow to senescence through the production of reactive oxygen species (ROS), further exacerbating heart function. Finally, they proposed that supplementation with *Lactobacillus* might be able to reduce the senescence of bone marrow mesenchymal cells, but this did not attenuate gut atrophy.

Combined it seems to be an interesting story and has the potential to be translated to the clinic which is very important, but the new mechanisms seem to have been described elsewhere, including a role for gut-bone marrow in cardiovascular disease, and a link between *Lactobacillus*, D-galactose, senescence and ROS. Moreover, importantly the authors did not measure heart

function in their models.

Major comments:

- 1- Did the CCHD patients have any lactose intolerance? Is this something that has been reported in the literature? Do they have gastro-intestinal issues if they have gut dystrophy?
- 2- Only 3 rats per group were used. Please show that this has enough statistical power. Based on our experience, a larger sample size would be required even in animal studies. Also what sex of rats were used? Both?
- 3- Because of reproducibility issues, in vitro studies should be ideally performed in at least triplicates in 3 independent experiments – the experiments reported seem to have been done in triplicates but only one time.
- 4- For the gut microbiome, please add what regions of 16S/18S/ITS genes were sequenced and the primers used.
- 5- For the statistical tests, it is not clear if multiple comparison adjustment was performed or not, so P used for significance should be smaller than the <0.05 described. This should have been done where there were more than 3 groups and for all metabolomics and gut microbiome data.
- 6- Is Figure 4 data from 16S or metagenomics? The reported 20,000 reads/sample in the rarefaction curve (Fig 4A) is quite low compared to recent studies where a depth of 50,000-100,000 would be analysed.
- 7- Figure 4F says the analysis show biomarkers between the two groups. I think this is an over-interpretation of this data as it only shows an association.
- 8- Figure 6: there is no significance described.
- 9- With supplementation of Lactobacillus, is there any data on heart function? Ultimately if the authors want to propose it as a new therapy this needs to be performed and conclusions about it being beneficial cannot be drawn without it.

Minor comments:

- 1- Check English grammar through the manuscript. Some sentences are also incomplete. Scientific writing also needs to be observed, the use of wording such as 'pretty lower' is not appropriate.
- 2- Some abbreviations are cited through the manuscript but only explained at the end in the methods. This is very confusing and difficult to follow. Please explain abbreviations at the first time they are shown in the text.
- 3- Real-time PCR should read as qPCR and not QPCR (page 23).
- 4- If paraffin-embedded samples were stained for H&E this is only histology and not immunohistochemistry.
- 5- Figures 1A, 3A, 5 and 6E: need to quantify each protein relative to b-actin, not only p16, and provide it in a graph with proper statistical analyses.
- 6- Figure 3: a two-way ANOVA with multiple comparison adjustment should have been performed, not t-test.
- 7- Firmicutes to Bacteroidetes ratio was not reported in the manuscript but it is referenced to in the discussion.

Response letter to reviewers

We appreciate the constructive comments and believe our manuscript has been improved significantly after addressing the comments. Please see below for our point-by-point responses to these comments.

Reviewers' comments:

Reviewer #1 (Remarks to the Author):

The manuscript by Xing et al. describes the hypoxia-induced changes to gut microbiota and bone marrow MSCs. They conclude that severe hypoxia induces gut microbiota dysbiosis and that this in turn leads to higher levels of D-galactose and subsequent MSC dysfunction and senescence. Using both patient data and a rat model, they suggest that introducing Lactobacillus can reduce D-galactose levels and rescue MSC function. The data are presented clearly and represent a thorough analysis of both human samples and rat tissue. The overall observations are very intriguing, but some conclusions seem over-interpreted, based on the data provided. It is also unclear from the manuscript why MSC function is the focus or read-out. The authors reference two papers suggesting that bone marrow stem cells can be used to treat myocardial infarction, but it seems very loosely connected to this study. Bone marrow contains hematopoietic, endothelial, and mesenchymal stem cells.

Thank you for the suggestions. Bone marrow-derived MSCs are well known for their prospective applications in stem cell therapy for cardiovascular diseases. Thus, our present research mainly focused on uncovering the underlying mechanism of MSC deficiency in CCHD patients and exploring some simple approaches for the amelioration of MSCs, both of which apparently benefit the implicated regenerative therapy with MSCs. Per your suggestion, we have replaced the references with new papers focusing on bone marrow-derived MSC therapy for heart disease. Furthermore, the mentioned inaccurate descriptions have been corrected in the revised manuscript.

Major.

1. It is not clear how Lactobacillus would regulate D-gal in the bone marrow. There are much higher levels of D-gal in the bone marrow- even non-Cy, as compared to the peripheral blood. What is the connection between the intestine and where Lactobacillus, presumably, is located, and the bone marrow? This is not clearly explained or discussed.

In our study, the amount of *lactobacillus* was markedly reduced in patients with CCHD, which correlated negatively with D-galactose accumulation in the peripheral blood. As a constituent of milk, lactose is the major nutrient for infants and young children. Lactose is degraded to glucose and galactose in the intestine and absorbed into the body. Most of the absorbed galactose is then degraded by the liver for energy production, while only a small amount is used for human early development ¹.

In the present study, the 16s results for patients with CCHD showed reduced *Lactobacillus* (one of the most important probiotic bacteria responsible for lactose and D-galactose metabolism in the intestinal tract), and furthermore, atrophied intestinal villus with impaired mucosal barrier fortification and reduced GALK1 expression in liver were observed in the rat hypoxia model. Therefore, we presume that the reduced *Lactobacillus*, atrophied intestinal villus and impaired liver function work together to contribute to the accumulation of D-galactose.

D-galactose would flow through the arteries to penetrate the cortical bone and medullary canal and then into the BM metaphyseal region and arterial capillaries. Based on the specified physiological structures, the metabolites would enter the BM niche and interact with MSCs. Several studies have shown that the metabolite distribution differs significantly but is correlated between bone marrow and peripheral blood, such as glucose² and lactate³. Our results also showed that the D-galactose concentration was significantly increased in BM compared with peripheral blood, in both the Cy group and the NCy group.

A series of enzymes that metabolize D-galactose are also present in BMSCs, but there were no significant changes between the NCy and Cy groups (Fig. S4B). Therefore, excessive D-galactose would trigger the production of ROS and ultimately induce premature senescence of MSCs (Fig. 3).

We have revised the discussion to clarify this issue per your suggestion.

2. Are there other reasons why there might be elevated D-gal concentrations in CCHD patients? Is D-gal on apoptotic cells? Are there more apoptotic cells in these patients?

To our knowledge, the regulation of the D-gal level is a relatively complicated process that is fine-tuned by both the intestine and liver. In the rat model of cyanosis, a detailed inspection of the process was conducted in our study. Apart from the remarkable decrease in *Lactobacillus* in the gut, our results suggested that decreased expression of liver GALK1 could contribute jointly to the D-gal elevation in CCHD patients. Nonetheless, gut dysbiosis was likely the main reason for the accumulation of D-gal since supplementation with *Lactobacillus* dramatically restored the D-gal level in rat models.

Indeed, D-gal has been reported to harbour a cytotoxic effect on cells, either for expediting senescence or inducing apoptosis⁴. It is worth noting that both cytotoxic effects depend largely on ROS through the oxidative metabolism of D-gal as well as advanced glycation end products⁵. It is well known that cells enter a state of senescence when the ROS level is sub-lethal⁶ or apoptosis after exposure to high levels of ROS⁷. Per your thoughtful suggestions, we have performed additional experiments to determine the level of apoptosis in the implicated BMSCs. Intriguingly, BMSCs from patients with CCHD exhibited much worse anti-apoptotic capability than those from non-cyanotic patients, indicative of the propensity for apoptosis in CCHD patients (Fig. S1C). Although a complete comprehension of the involved molecular mechanisms between cellular senescence and apoptosis remains to be achieved, it is believed that ROS levels play a pivotal role in balancing the two physiological processes⁸. Moreover, ageing may predispose related cells to apoptosis through enhanced mitochondrial dysfunction despite some controversies^{9,10}. Herein, the observed premature senescence of deficient BMSCs from CCHD patients appeared to be the primary toxic effect of D-gal in our study due to the relatively low level of accumulated D-gal in contrast to that used in other reports. With excessive exposure to continuously increasing ROS, some BMSCs or senescent BMSCs likely undergo a stage of disruption of cellular function and membrane integrity and, in turn, are induced to undergo apoptosis. The underlying mechanism responsible for this

process clearly merits further characterization.

We have revised the Discussion to clarify this issue per your suggestion.

3. One disconnect is why BM MSCs are important in heart disease? Are there confounding impacts of CCHD on hematopoiesis, for example?

Thank you for the good suggestions. BMSCs have been widely reported to be a promising candidate cells for cell-based therapy for heart disease¹¹⁻¹³. For children with CCHD, BMSCs are also a widely applied cell resource for the fabrication of tissue engineering patches or conduits¹⁴. In this regard, it appears to be extremely important to elucidate the functional integrity of BMSCs in CHD patients. The present study showed the deficiency of BMSCs in cyanotic CHD patients compared with non-cyanotic ones and further investigated the underlying mechanism, which might facilitate the development of future therapies for CCHD.

With regard to the impact of CCHD on haematopoiesis, previous studies have shown that hypoxia in CCHD likely stimulates the increase in haemoglobin through increases in EPO, which is generally considered to be a compensatory regulation. Interestingly, Randa et al.¹⁵ recently found that the elevated immature platelet fraction in CCHD patients may be indicative of peripheral platelet destruction. Apparently, haematopoiesis has been perturbed to some extent in CCHD patients. Given that BMSCs and HSCs share the same microenvironment in bone marrow, it is conceivable that the accumulated D-gal in CCHD patients influences the function of HSCs, and even other progenitor cells residing in the bone marrow. We are interested in exploring the dynamic changes in HSCs and likely malfunction of haematopoiesis

in CCHD patients.

We have described this issue as a limitation of study in the revised manuscript.

4. Legends in some figures are almost impossible to read due to the very small font size (Figure 4, for example).

Your points are well taken. All figures have been re-edited in the revised manuscript.

Minor

1. The entire manuscript contained grammatical mistakes. Please review carefully.

We very much appreciate your suggestion. We have revised the manuscript carefully and corrected the grammatical mistakes.

2. The histograms are not labeled clearly in Supp. Figure 1.

We have relabelled the histograms in the revised manuscript.

3. The first part of the results section where patient enrollment is described is confusing- particularly the last statement which states "among them"- it is not clear who is "them".

Thank you for the suggestion. We have replaced "them" with "the 90 patients" in the revised version of the manuscript.

Reference

1. Coelho, A.I., Berry, G.T. & Rubio-Gozalbo, M.E. Galactose metabolism and health. *Current opinion in clinical nutrition and metabolic care* 18, 422-427 (2015).
2. Tiziani, S., *et al.* Metabolomics of the tumor microenvironment in pediatric acute lymphoblastic leukemia. *PLoS One* 8, e82859 (2013).
3. Suda, T., Takubo, K. & Semenza, G.L. Metabolic regulation of hematopoietic stem cells in the hypoxic niche. *Cell Stem Cell* 9, 298-310 (2011).
4. Li, N., *et al.* D-galactose induces necroptotic cell death in neuroblastoma cell lines. *Journal of cellular biochemistry* 112, 3834-3844 (2011).
5. Yu, Y., *et al.* Fibroblast growth factor (FGF21) protects mouse liver against D-galactose-induced oxidative stress and apoptosis via activating Nrf2 and PI3K/Akt pathways. *Molecular and cellular biochemistry* 403, 287-299 (2015).
6. Lu, T. & Finkel, T. Free radicals and senescence. *Experimental cell research* 314, 1918-1922 (2008).
7. Cao, C., *et al.* AMP-activated protein kinase contributes to UV- and H₂O₂-induced apoptosis in human skin keratinocytes. *The Journal of biological chemistry* 283, 28897-28908 (2008).

8. Fulle, S., *et al.* Stem cell ageing and apoptosis. *Current pharmaceutical design* 18, 1694-1717 (2012).
9. Lee, H.C. & Wei, Y.H. Oxidative stress, mitochondrial DNA mutation, and apoptosis in aging. *Experimental biology and medicine* 232, 592-606 (2007).
10. Chistiakov, D.A., Sobenin, I.A., Revin, V.V., Orekhov, A.N. & Bobryshev, Y.V. Mitochondrial aging and age-related dysfunction of mitochondria. *BioMed research international* 2014, 238463 (2014).
11. Hare, J.M., *et al.* A randomized, double-blind, placebo-controlled, dose-escalation study of intravenous adult human mesenchymal stem cells (prochymal) after acute myocardial infarction. *Journal of the American College of Cardiology* 54, 2277-2286 (2009).
12. Chen, S.L., *et al.* Effect on left ventricular function of intracoronary transplantation of autologous bone marrow mesenchymal stem cell in patients with acute myocardial infarction. *The American journal of cardiology* 94, 92-95 (2004).
13. Shin'oka, T., *et al.* Midterm clinical result of tissue-engineered vascular autografts seeded with autologous bone marrow cells. *J Thorac Cardiovasc Surg* 129, 1330-1338 (2005).
14. Drews, J.D., Miyachi, H. & Shinoka, T. Tissue-engineered vascular grafts for congenital cardiac disease: Clinical experience and current status. *Trends in cardiovascular medicine* 27, 521-531 (2017).
15. Matter, R.M., *et al.* Determinants of platelet count in pediatric patients with congenital cyanotic heart disease: Role of immature platelet fraction. *Congenital heart disease* (2017).

Reviewer #2 (Remarks to the Author):

*The study by Zhang and colleagues presents a novel angle that could be involved in cyanotic congenital heart disease, proposing a hypoxia-gut-bone marrow link and feedback loop. They propose that hypoxia because of a mal-functioning heart modulates the gut microbiota and reduces *Lactobacillus*, thus resulting in an accumulation of *D*-galactose. This in turn drives cells in the bone marrow to senescence through the production of reactive oxygen species (ROS), further exacerbating heart function. Finally, they proposed that supplementation with *Lactobacillus* might be able to reduce the senescence of bone marrow mesenchymal cells, but this did not attenuate gut atrophy.*

*Combined it seems to be an interesting story and has the potential to be translated to the clinic which is very important, but the new mechanisms seem to have been described elsewhere, including a role for gut-bone marrow in cardiovascular disease, and a link between *Lactobacillus*, *D*-galactose, senescence and ROS. Moreover, importantly the authors did not measure heart function in their models*

Thank you for the thoughtful suggestions. There is no doubt that improved heart function is an important parameter for evaluating the efficacy of heart cell therapy. However, in the present study, we wanted to emphasize how to rescue the premature senescence of BMSCs in children with CCHD. In the rat hypoxia model, compared with the myocardial infarction model, impaired heart function could not be observed during the experimental period (3 weeks) (the data have been supplied in the Supplemental files). Therefore, we would like to perform a new study in the future to investigate the effect of transplantation of rescued or non-rescued BMSCs on heart function with a prolonged experimental period. In the clinical setting, in CCHD patients, the purpose of cell-based regenerative therapy is to utilize cell infusion to prevent the deterioration of heart function or cell seeding to fabricate a tissue engineered conduit. Our findings will also be helpful for the optimization of cell resources for tissue engineering for CCHD patients.

Major comments:

1- Did the CCHD patients have any lactose intolerance? Is this something that has been reported in the literature? Do they have gastro-intestinal issues if they have gut dystrophy?

Until now, no studies have indicated the correlation between CCHD patients and lactose intolerance. Lactose intolerance is a condition in which affected people have symptoms due to the decreased ability to digest lactose. This is mainly caused by a relative or absolute absence of lactase, small bowel injury or congenital lactase deficiency.^{16,17} A few studies have reported that approximately 20% of Asian children younger than 5 years of age have evidence of lactose intolerance.^{18,19} In the clinical setting, the diagnosis of lactose intolerance depends on symptoms following lactose consumption. The symptoms of lactose intolerance included diarrhoea, nausea, flatulence, and/or bloating. In our study, we excluded all children with intestinal malformations or with a history of recent serious abdominal symptoms. Lactose-free formulas were used in the lactose intolerance treatment. To further clarify this confounding factor, we reviewed the milk type used for the enrolled patients. We found that 4 patients in the NCy group were given formula milk, as compared to 5 patients in the Cy group ($p = 0.725$). In the rat hypoxia model, we also confirmed that hypoxia did not affect the intestinal lactase enzyme activity (Fig. 6D). Taken together, we did not observe correlation between patients with CCHD and lactose intolerance. Therefore, we presumed that the presence of lactose intolerance did not affect the conclusion.

Our results revealed intestinal dystrophy in the hypoxic rat model. Children with CCHD may have gastro-intestinal issues. Neonates with CCHD have a high incidence of necrotizing enterocolitis²⁰. Because all the patients in the present study were beyond the first month after birth, we did not find any serious abdominal symptom in the enrolled CCHD patients. However, poor body weight could be observed in CCHD patients, which would have been caused, in part, by gut dystrophy and potentially poor abnormal neurodevelopment.

2- Only 3 rats per group were used. Please show that this has enough statistical power. Based

on our experience, a larger sample size would be required even in animal studies. Also what sex of rats were used? Both?

Thank you for your important suggestions. We have increased the sample size and repeated the animal study. Now each group had 9 rats(6+3 / group). The results were consistent with the previous data, and thus, we have revised the data in the manuscript. To reduce the size of figures, we still present 3/group for the western blotting. And here is the western blotting results of new samples:

All the experimental animals were young male rats. Because all the patients enrolled in the study were very young children, we believe that gender may not be a major factor in the intestinal microbiota disorders observed under hypoxia.

3- Because of reproducibility issues, in vitro studies should be ideally performed in at least triplicates in 3 independent experiments – the experiments reported seem to have been done in triplicates but only one time.

Thank you for the thoughtful suggestions. In fact, all the experiments were performed in triplicate in 3 independent experiments, but we present only one of them. We have increased the sample size and repeated the animal study to confirm the conclusion.

4- For the gut microbiome, please add what regions of 16S/18S/ITS genes were sequenced and the primers used.

Your point is well taken. We have now provided the sequenced regions and primers in the Materials and Methods.

5- For the statistical tests, it is not clear if multiple comparison adjustment was performed or not, so P used for significance should be smaller than the <0.05 described. This should have been done where there were more than 3 groups and for all metabolomics and gut microbiome data.

Thank you for the suggestions. All statistical analyses were repeated with multiple comparison adjustment as described in the revised manuscript. Now the adjustment P values were presented for more than 3 groups and metabolomics and gut microbiome data .

6- Is Figure 4 data from 16S or metagenomics? The reported 20,000 reads/sample in the rarefaction curve (Fig 4A) is quite low compared to recent studies where a depth of 50,000-100,000 would be analysed.

Yes, the data from figure 4 were based on the results of 16s and metagenomics analyses. Since the study was launched in 2013, technical limits restricted it to 30,000 reads/sample. Per your requirement, we have re-sequenced the corresponding samples preserved in liquid nitrogen, and the results are presented in the new figure 4. We have also revised the corresponding methods section.

7- Figure 4F says the analysis show biomarkers between the two groups. I think this is an over-interpretation of this data as it only shows an association.

Thank you for the suggestion. We have changed “biomarkers” to “association” in the revised manuscript.

8- Figure 6: there is no significance described.

Thank you for the important suggestions. We have added the significance in the revised manuscript.

9- With supplementation of Lactobacillus, is there any data on heart function? Ultimately if the authors want to propose it as a new therapy this needs to be performed and conclusions about it being beneficial cannot be drawn without it.

Per your suggestions, we have performed additional experiments to examine the heart function of the rats (Fig. S5C). We have also listed this issue as a study limitation and revised the conclusion per your suggestion.

Minor comments:

1- Check English grammar through the manuscript. Some sentences are also incomplete. Scientific writing also needs to be observed, the use of wording such as 'pretty lower' is not appropriate.

We very much appreciate your suggestion. We have asked a native English speaker to correct the grammatical mistakes and inappropriate language.

2- Some abbreviations are cited through the manuscript but only explained at the end in the

methods. This is very confusing and difficult to follow. Please explain abbreviations at the first time they are shown in the text.

Thank you for the suggestions. We have defined the abbreviations the first time they mentioned in the revised manuscript.

3- Real-time PCR should read as qPCR and not QPCR (page 23).

Thank you for the suggestions. This has been corrected in the revised methods.

4- If paraffin-embedded samples were stained for H&E this is only histology and not immunohistochemistry.

We have corrected this in the revised methods.

5- Figures 1A, 3A, 5 and 6E: need to quantify each protein relative to b-actin, not only p16, and provide it in a graph with proper statistical analyses.

Your point is well taken, and the graph is now provided in the revised manuscript.

6- Figure 3: a two-way ANOVA with multiple comparison adjustment should have been performed, not t-test.

Your points are well taken, and we have performed the statistical analysis in the revised manuscript.

7- Firmicutes to Bacteroidetes ratio was not reported in the manuscript but it is referenced to in the discussion.

We have added the Firmicutes-to-Bacteroidetes ratio in the Results section.

Reference

16. American Academy of Pediatrics Committee on Nutrition: Practical significance of lactose intolerance in children: supplement. *Pediatrics* **86**, 643-644 (1990).
17. The practical significance of lactose intolerance in children. *Pediatrics* **62**, 240-245 (1978).
18. Woteki, C.E., Weser, E. & Young, E.A. Lactose malabsorption in Mexican-American children. *The American journal of clinical nutrition* **29**, 19-24 (1976).
19. Heyman, M.B. & Committee on, N. Lactose intolerance in infants, children, and adolescents. *Pediatrics* **118**, 1279-1286 (2006).
20. Ellis, C.L., Rutledge, J.C. & Underwood, M.A. Intestinal microbiota and blue baby syndrome: probiotic therapy for term neonates with cyanotic congenital heart disease. *Gut Microbes* **1**, 359-366 (2010).

Reviewers' comments:

Reviewer #1 (Remarks to the Author):

The revised version is much improved. Some axes are still problematic, however, such as Figure 1D.

The issues raised were addressed through editorial modifications.

Reviewer #2 (Remarks to the Author):

The manuscript has been significantly improved. It would have been helpful to see a version of the manuscript with the changes marked. Where possible, most comments were addressed.

In the light of a recent paper published in Nature (<https://www.nature.com/articles/nature24628>) showing a link between *Lactobacillus* and blood pressure, did the authors observe any association in their cohort?

I don't work with hypoxia models so I cannot comment on the model itself. The hypothesis behind their manuscript was that CCDH caused hypoxia, which then caused changes in the gut microbiota, resulting in bone marrow senescence, which in turn exacerbated heart disease. Based on their results from echocardiography this did not seem to be the case as there was no change in heart function.

Reviewer#3 (Remarks to the Author):

The work of Xing et al describes the association between hypoxia in cyanotic heart disease with perturbed gut microbiota, circulating metabolites and bone marrow mesenchymal stem cell (MSC) senescence. The work is interesting and in general well performed, but the link between MSC function and cardiovascular disease is poorly addressed and in fact not important for the point the manuscript explores, which is the link between hypoxia, gut dysgenesis and MSC senescence. To address this concern, comments follow below:

1. The animal model used in the study is not a CHD model, but a hypoxic model. Therefore, any conclusions about the role the microbiome/MSCs play in CHD are not appropriate and should be removed from the manuscript. In fact, CHD seems to be irrelevant for the described phenotype, even though it is the cause of cyanosis in patients.

On the same topic, the Echo data presented in the paper has no value for interpretation of the results, because 1. This model is not a CHD model, as previously stated; 2. There is no evidence that endogenous MSCs populate the heart of patients or animal models with CHD or any other cardiovascular disease, as a matter of fact. In fact, most murine studies show that cardiac fibroblasts are the major source of cells in diseased hearts. The authors should also be aware of the consensus report on cardiomyocyte regeneration (*Circulation*. 2017;136:680–686. DOI: 10.1161/CIRCULATIONAHA.117.029343), which addresses the contentious role of cardiac progenitor cells and MSCs in heart disease. Most studies seem to suggest that the role of MSCs is a paracrine one, instead of a regenerative one. The presentation of the Echo data could also be improved in case this dataset remains in the manuscript. The number of animals used in the study should be included, and the graph should show each individual datapoint in the bar, as heart function is subjected to strong biological variation.

2. Due to point 1, I suggest the authors change the title to 'Hypoxia driven by cyanotic heart disease induces senescence of bone marrow mesenchymal stem cells via altered gut microbiota', or something similar that clarifies the findings are not necessarily related to CHD, but to hypoxia.
3. The authors cannot exclude the fact that gut dysgenesis independent of the microbiome is also a cause for the observed phenotype either in patients or in the animal model, especially considering the observed gut atrophy shown in Figure 6. Therefore, this hypothesis should be also carefully laid out in the discussion section.
4. The treatment using *Lactobacillus* (Figure 6) to rescue MSC senescence is interesting, but another contentious point. Many are of the opinion that probiotics do not repopulate the gut flora or provide long-term beneficial effects to individuals. Have the authors looked at chronic situations in which *Lactobacillus* is removed from the diet of hypoxic animals to see if these effects are permanent or simply acute?

Minor point:

1. On the introduction, the authors claim that 'Surgical corrections are the most effective treatment for patients with CCHD. Sometimes, primary repair is possible, but replacement grafts are more often required.' This point is unclear, as MSC based grafts are not currently used in the clinic to treat CHD. The authors should elaborate on the kind of replacement grafts currently used in corrective surgeries or remove this point, so that readers are not misguided in regards to the use of MSC-derived explants in CHD.
2. Still in the introduction, the authors reference 2 studies (Refs 6 and 7) suggesting the use of stem cell therapy to treat congenital heart disease. Again, this point can be misleading to readers, as there is no proof of the efficacy of such treatments to correct CHD, which is mostly accompanied by anatomical defects not feasible of correction using autologous stem cell injections. The authors should also explain this statement.
3. A third comment on the introduction states that 'Accumulating evidence has supported the hypothesis that gut-BM communication is involved in the pathogenesis of cardiovascular diseases'. The statement per se may be true, but the current work does not support a role for the gut-BM in cardiovascular disease, but points to hypoxia driven by CHD as a cause for gut-BM disturbances.
4. A last point I would like the authors to consider is that although in vitro assays are classic measurements of the differentiation power of MSCs, these properties are not necessarily desirable for a cell composing the heart. In fact, calcifications and adipogenesis are detrimental to heart function, and shouldn't be promoted as a beneficial potential for MSCs in heart disease.

Response letter to reviewers

We appreciate the constructive comments and believe our manuscript has been improved significantly after addressing the comments. Please see below for our point-by-point responses to these comments.

Reviewers' comments:

Reviewer #1 (Remarks to the Author):

The revised version is much improved. Some axes are still problematic, however, such as Figure 1D.

The issues raised were addressed through editorial modifications.

Thank you for the suggestion. All figures have been re-edited in the revised manuscript to meet the requirements of the journal.

Reviewer #2 (Remarks to the Author):

The manuscript has been significantly improved. It would have been helpful to see a version of the manuscript with the changes marked. Where possible, most comments were addressed.

In the light of a recent paper published in Nature

*(<https://www.nature.com/articles/nature24628>) showing a link between *Lactobacillus* and blood pressure, did the authors observe any association in their cohort?*

Thank you for the thoughtful suggestion. Since the CCHD patients has relatively lower level of *Lactobacillus*, we reviewed our cohort and did not find altered blood pressure in CCHD group. Wilck and colleagues' data showed high the salt consumption reduced survival of *Lactobacillus*, increased Th17 cells and blood pressure. Their study concerned healthy adults with normal blood pressure and a high salt diet for a relatively short time. As we know, blood pressure is regulated by many factors, such as kidneys, sympathetic nervous system, vasculature, immune system and so on. Our enrolled patients were children. Neither the gut Microbiota¹⁻³ nor the immune system⁴ share the same between

children and adult. Therefore, in our cohort, we didn't observe the altered blood pressure in the CCHD group.

We have listed this issue in the study limitation.

I don't work with hypoxia models so I cannot comment on the model itself. The hypothesis behind their manuscript was that CCHD caused hypoxia, which then caused changes in the gut microbiota, resulting in bone marrow senescence, which in turn exacerbated heart disease. Based on their results from echocardiography this did not seem to be the case as there was no change in heart function.

Thank you for the question.

It could be some misunderstandings towards this study because of our misleading descriptions in certain respects.

The present study explored the down-stream effects of hypoxia driven by CCHD, and the established rat hypoxia model help to elucidate the underlying mechanism. We have reorganized some parts of writings to focus this work on the relationship between chronic hypoxia and BMSCs function, not only limited in the field of cardiovascular disease. Indeed, in the clinical setting, the heart function of CCHD patient could maintained normal for a long time. Therefore, we did not observe any impaired heart function in either the hypoxic rat or the CCHD patients. The impaired heart function will occur in the end-stage of CCHD if the patients didn't receive the appropriate surgical repair.

Given this, it is proposed that the poor heart function of CCHD might not be imputable to BMSCs deficiency caused by hypoxia. Additionally, a naturally direct link of BMSCs to heart function could not be provided by literatures to date.

Reviewer#3 (Remarks to the Author):

1. The animal model used in the study is not a CHD model, but a hypoxic model.

Therefore, any conclusions about the role the microbiome/MSCs play in CHD are not appropriate and should be removed from the manuscript. In fact, CHD seems to be irrelevant for the described phenotype, even though it is the cause of cyanosis in patients.

Thanks so much and we cannot agree more with you. We have already removed the inappropriate descriptions in the manuscript per your suggestion. Actually, both the CCHD and rat model established here are under chronic hypoxia. What we want to explore is the relationship between chronic hypoxia and BMSCs function and to address the underlying mechanisms mediated by gut dysbiosis, both of which appear to benefit the related medical cure. Obviously, the observed phenotype of BMSCs in CCHD patients is merely linked to the pathological hypoxia driven by CCHD and the subsequent experiments performed on hypoxia rat model also support this.

On the same topic, the Echo data presented in the paper has no value for interpretation of the results, because 1. This model is not a CHD model, as previously stated; 2. There is no evidence that endogenous MSCs populate the heart of patients or animal models with CHD or any other cardiovascular disease, as a matter of fact. In fact, most murine studies show that cardiac fibroblasts are the major source of cells in diseased hearts. The authors should also be aware of the consensus report on cardiomyocyte regeneration (Circulation. 2017; 136: 680–686. DOI: 10.1161/CIRCULATIONAHA.117.029343), which addresses the contentious role of cardiac progenitor cells and MSCs in heart disease. Most studies seem to suggest that the role of MSCs is a paracrine one, instead of a regenerative one. The presentation of the Echo data could also be improved in case this dataset remains in the manuscript. The number of animals used in the study should be included, and the graph should show each individual data point in the bar, as heart function is subjected to strong biological variation.

Considering the question proposed in point 1, we realized that the Echo data presented here indeed seemed to be valueless in this study. And we have deleted it in the revision.

As you mentioned above, the contentious role of MSCs in heart disease has been widely reported (Circulation. 2017; 136: 680–686. DOI: 10.1161/CIRCULATIONAHA.117.029343). Meanwhile, it is noted that the medical application and functionality of BMSCs in heart disease also have been suggested, no matter based on regenerative or paracrine one. Given this, it is necessary to figure out the BMSCs function and related mechanisms in heart disease. Additionally, we must reiterate the goal of our work aiming to investigate how the hypoxia occurred in CCHD patients jeopardizes the BMSCs, which may provide some insights to the clinical therapy, not only for CCHD, but also for other analogous diseases.

2. Due to point 1, I suggest the authors change the title to 'Hypoxia driven by cyanotic heart disease induces senescence of bone marrow mesenchymal stem cells via altered gut microbiota', or something similar that clarifies the findings are not necessarily related to CHD, but to hypoxia.

Thank you for your suggestion, we have changed the title in the revised manuscript.

3. The authors cannot exclude the fact that gut dysgenesis independent of the microbiome is also a cause for the observed phenotype either in patients or in the animal model, especially considering the observed gut atrophy shown in Figure 6. Therefore, this hypothesis should be also carefully laid out in the discussion section.

Thank you for the suggestion, this part has been added in Discussion section of the revised manuscript. We have highlighted the changes as the follows:

'Lactose is also degraded by lactase, which is secreted by the intestinal gland around the brush border area. We did not observe impaired lactase activity in the intestine in the rat hypoxia model. Likewise, Intestinal mucosa atrophy was noted in our model, which has been previously observed both in hypoxia rat⁵ and patients with chronic obstructive pulmonary disease⁶, indicating impaired mucosal barrier fortification and thus the entry of more bacterial metabolites into the circulation⁵. Given this case, we cannot exclude the fact that gut dysgenesis independent of microbiome is also a cause for

the observed phenotype. We found that the replenishment of hypoxic rats with *Lactobacillus* could fully restore the deficient BMSCs, while the intestinal mucosa atrophy did not fully recover even after probiotic therapy. Therefore, the alterations of the gut microbial communities might be the main reason for the deficient BMSCs, and complete surgical repair of cardiac anomalies is warranted to relieve hypoxia and restore the integrity of the mucosal barrier.'

4. The treatment using *Lactobacillus* (Figure 6) to rescue MSC senescence is interesting, but another contentious point. Many are of the opinion that probiotics do not repopulate the gut flora or provide long-term beneficial effects to individuals. Have the authors looked at chronic situations in which *Lactobacillus* is removed from the diet of hypoxic animals to see if these effects are permanent or simply acute?

Thanks for your suggestions. This issue is very important for the clinical translation of probiotic therapy. We have performed the additional experiment as your suggestion. As shown in the new Fig S7, we found that when the *Lactobacillus* was removed from the diet of hypoxic rats, the BMSCs showed premature senescence again. It means the continues supplementary would be necessary before the hypoxia status is corrected.

Figure S7. Continues supplementary with *Lactobacillus* is necessary for recovery of deficient BMSCs. (A)

Rats were divided randomly into three groups (n=6 per group) and housed in the hypoxic chamber for 2 weeks, then supplied with *Lactobacillus* or saline for 3 weeks (-/- group, supplied with saline for entire 3 weeks; +/- group, *Lactobacillus* 1 week followed by saline 2 weeks; +/+ group, supplied with *Lactobacillus* for entire 3 weeks). Stools were subjected to qPCR to assess the *Lactobacillus* content. **(B)** Peripheral blood samples from the rats described in S7A were obtained, and the D-galactose concentrations were measured. **(C)** BMSCs were isolated and cultured to passage 3. SA- β -gal activity was assessed. Representative images are shown (upper). The ratio of β -galactosidase-positive cells were calculated. The data shown are the mean \pm SD, and statistical significance was analysed by one-way ANOVA followed by Tukey-Kramer multiple comparisons. a $P < 0.05$ compared with -/- group. b $P < 0.05$ compared with +/+ group.

We have added the new data in the supplement files in the revised manuscript.

Minor point:

1. On the introduction, the authors claim that ‘Surgical corrections are the most effective treatment for patients with CCHD. Sometimes, primary repair is possible, but replacement grafts are more often required.’ This point is unclear, as MSC based grafts are not currently used in the clinic to treat CHD. The authors should elaborate on the kind of replacement grafts currently used in corrective surgeries or remove this point, so that readers are not misguided in regards to the use of MSC-derived explants in CHD.

Thank you for the suggestion. I’m sorry for the obscure and misleading expression and this part has been removed in the revised manuscript.

2. Still in the introduction, the authors reference 2 studies (Refs 6 and 7) suggesting the use of stem cell therapy to treat congenital heart disease. Again, this point can be misleading to readers, as there is no proof of the efficacy of such treatments to correct CHD, which is mostly accompanied by anatomical defects not feasible of correction using autologous stem cell injections. The authors should also explain this statement.

Thank you for the reminding. The inappropriate statement has been modified.

3. A third comment on the introduction states that ‘Accumulating evidence has supported the hypothesis that gut-BM communication is involved in the pathogenesis of cardiovascular diseases’. The statement per se may be true, but the current work does

not support a role for the gut-BM in cardiovascular disease, but points to hypoxia driven by CHD as a cause for gut-BM disturbances.

Thanks for the suggestion, I have made the corresponding adjustments in the revised manuscript in the light of your proposal.

4. A last point I would like the authors to consider is that although in vitro assays are classic measurements of the differentiation power of MSCs, these properties are not necessarily desirable for a cell composing the heart. In fact, calcifications and adipogenesis are detrimental to heart function, and shouldn't be promoted as a beneficial potential for MSCs in heart disease.

Thank you for the suggestion. This part has been amended in the manuscript to highlight the multifunction of BMSC rather than its application in heart disease treatment.

Reference

1. Ellis, C.L., Rutledge, J.C. & Underwood, M.A. Intestinal microbiota and blue baby syndrome: probiotic therapy for term neonates with cyanotic congenital heart disease. *Gut microbes* 1, 359-366 (2010).
2. Koenig, J.E., *et al.* Succession of microbial consortia in the developing infant gut microbiome. *Proceedings of the National Academy of Sciences of the United States of America* 108 Suppl 1, 4578-4585 (2011).
3. Yatsunenko, T., *et al.* Human gut microbiome viewed across age and geography. *Nature* 486, 222-227 (2012).
4. Simon, A.K., Hollander, G.A. & McMichael, A. Evolution of the immune system in humans from infancy to old age. *Proceedings. Biological sciences* 282, 20143085 (2015).
5. Xu, C.L., *et al.* Protective effect of glutamine on intestinal injury and bacterial community in rats exposed to hypobaric hypoxia environment. *World journal of gastroenterology* 20, 4662-4674 (2014).
6. Fedorova, T.A., *et al.* [The stomach and duodenum condition in patients with chronic obstructive lung diseases]. *Klinicheskaia meditsina* 81, 31-33 (2003).

REVIEWERS' COMMENTS:

Reviewer #3 (Remarks to the Author):

I am satisfied with the modifications made by the authors